

# Implementation of the Bessel's method for solar eclipses prediction in the WRF-ARW model

A. Montornès[1], B. Codina[1], J. W. Zack[2], and Y. Sola[1]

[1]Department of Astronomy and Meteorology, University of Barcelona, Barcelona, Spain
[2]MESO Inc., Troy, USA

*Correspondence to:* A. Montornès (amontornes@am.ub.es)

**Abstract.**

Solar eclipses are predictable astronomical events that abruptly reduce the incoming solar radiation into the Earth's atmosphere, which frequently result in non-negligible changes in meteorological fields. The meteorological impacts of these events have been analyzed in many studies since the late 1960s. The recent growth in the solar energy industry has greatly increased

the interest in adding additional detail to the modeling of solar radiation variations in Numerical Weather Prediction (NWP) models for use in solar resource assessment and forecasting applications. The recent partial and total solar eclipses that occurred in USA (October 23, 2014) and Europe (March 20, 2015), respectively, are showing the necessity for including these astronomical events on the current solar parameterizations, beyond the purely meteorological interest.

Although some studies added solar eclipse episodes within NWP codes in the 1990s and 2000s, they used eclipse param-
eterizations designed for a particular case of study. In contrast to these earlier implementations, this paper documents a new package for the Weather Research and Forecasting - Advanced Research (WRF-ARW) model that can simulate any partial, total or hybrid solar eclipse for the period 1950 to 2050 and is also extensible to a longer period. The algorithm computes analytically the trajectory of the Moon's shadow and the degree of obscuration of the solar disk at each grid-point of the domain based on the Bessel's method and the Five Millennium Catalog of Solar Eclipses provided by NASA, with a negligible
computational time. Then, the incoming radiation is modified accordingly at each grid-point of the domain.

This contribution is divided in two parts. First, we present a description of the implementation of the Bessel's method within the WRF-ARW model together with a validation for the period 1950-2050 of all solar eclipse trajectories with respect to NASA values. Second, we analyze the model response in four total solar eclipse episodes: 1994-11-03 (South America), 1999-08-11 (Europe), 2006-03-29 (North Africa) and 2009-07-22 (Eastern Asia). The second part includes a validation of the simulated
global horizontal irradiance (GHI) with measurement data from selected Baseline Surface Radiation Network sites within the area affected by each event as well as an analysis of the impact of the GHI changes in surface temperature and wind speed.

## 1   Introduction

Solar eclipses are predictable astronomical events that reduce momentarily the incoming radiation to the Earth's atmosphere, inducing a significant change on the meteorological fields. The impact of the shadow of the Moon on the Earth's atmosphere





has awakened the interest of many scientists since the second part of the $20^{th}$ century. For example, the region under the shadow of an eclipse experiences similar surface and planetary boundary layer (PBL) processes to those that occur at sunrise and sunset but they are more abruptly and on a shorter time-scale (Anderson, 1999). This provides a unique opportunity to analyze these processes.

The first modern studies related to the relationship between the atmosphere and the solar eclipses appeared in the late 1960s mainly focused on the ozone variations and their impact on the stratosphere and mesosphere (Bojkov, 1968; Randhawa, 1968; Ballard and Valenzuela, 1969).

At the beginning of the 1970s, Chimonas and Hines (1970, 1971) suggested that the cooling produced by the lunar shadow, crossing the atmosphere of the Earth at supersonic speeds, should produce gravity waves at the upper layers measurable as
surface pressure fluctuations. Based on this discussion, many studies appeared trying to detect these waves such as in Davis and Rosa (1970); Anderson (1972); Chimonas (1973) or later publications as Fritts and Luo (1993); Altadill et al. (2001); Zerefos (2007).

Although several early studies (e.g. Stewart and Rouse (1974); Antonia (1979)) examined the impact of solar eclipses on surface processes, it was not until the late 1990s and early 2000s that the focus shifted to the variations of temperature, humidity,
wind speed, turbulence and atmospheric chemistry. Fernández et al. (1993b, a) analyzed the variations produced by the total eclipse of July 11, 1991 on different meteorological fields using a set of surface stations and radio-soundings for different sites in Costa Rica. The first study concerned the impact on the global horizontal irradiance (GHI) measurements while the second one analyzed the effect on temperature, humidity and wind speed, at the ground and at the free atmosphere. Fernández et al. (1993a) observed negative temperature deviations ranging from 2 to 5º C reaching the minimum value between 10–30 min
after the maximum obscuration of the solar disk. Surface wind speed experienced a noteworthy reduction some minutes after reaching the lowest temperature in those sites not dominated by the large scale patterns. In the free atmosphere, the highest temperature and wind speed variations were observed at ∼13 km (i.e. 175 hPa) with thermal differences from -2 to -6º C and a high deviation in wind direction compared with soundings on similar days. Other authors as Fernández and Hidalgo (1996); Segal and Turner (1996); Anderson (1999), among others reported similar results focusing on surface temperature. Meanwhile,
Eaton et al. (1997) examined the effects of a solar eclipse on the PBL using for the study the episode of May 10, 1994. The analysis showed a clear impact on the heat exchange (sensible and latent), a reduction of the turbulence processes and a significant negative deviation on the refractive index structure parameter. Moreover, using a frequency-modulated, continuous-wave (FM-CW) radar operating at 2.9 GHz, Eaton et al. (1997) reported the development of Kelvin–Helmholtz waves during the eclipse.

The most meteorologically analyzed eclipse event is the total solar eclipse that occurred over Europe on August 11,1999. The expanded use of mesoscale NWP models during the late 1990s along with the dense network of weather stations across Europe facilitated a large number of publications on this event. In general, these studies focused on the impact on near-surface variables. For example, Hanna (2000) analyzed the measured variation at stations across the United Kingdom while Aplin and Harrison (2003) provided a broader scale analysis of the variations across the continent. Other relevant studies in other areas
of interest were performed by Abram et al. (2000); Zerefos et al. (2001); Anfossi et al. (2004), among others. Abram et al.



(2000) measured the effect of the solar reduction on the tropospheric chemistry, particularly on the hydroxyl radical and ozone in England. Zerefos et al. (2001) examined the induced thermal fluctuations in the ozone layer, ionosphere and troposphere at stations in the Balkans observing the existence of dominant oscillations in the parameters related to the ionosphere and the ozone layer. Anfossi et al. (2004) used a 3-axis propeller anemometer (Gill-type) and a fast response temperature sensor

in a mast located in France for measuring the turbulence variation during the eclipse. In that study, they documented a rapid turbulent kinetic energy decay in time.

More recently, other studies such as Founda and Melas (2007); Gerasopoulos et al. (2008) have been focused on the total solar eclipse occurred in March 29, 2006 over Eastern Europe and particularly, in Greece. Subrahamanyam and Anurose (2011a) analyzed the behavior of the atmospheric surface layer comparing the eclipse observations for the event in January 15,

2010 with similar measurements recorded on non-eclipse cloudless days in India used as a baseline.

Given the low frequency of solar eclipses in regions with meteorological stations, atmospheric models are suitable tools for analyzing the response of the atmosphere during a solar eclipse episode. The first studies with models appear during the 1990s. Segal and Turner (1996) used a boundary layer version of the model described in Arritt (1989) in order to evaluate the spatial and temporal effects on shelter temperature using the total solar eclipse of May 10, 1994. The general features of the eclipse

were quantified using data from the ephemerides and then refined by direct computation of the sun/moon geometry based on standard methods of celestial mechanics with a suitable accuracy.Gross and Hense (1999) presented a NWP model study using the Deutschland-Modell (DM) from the German Weather Service (DWD) for analyzing the meteorological effects of the August 11, 1999 eclipse. In this case, the episode was parameterized in terms of the shadow's trajectory and approximating the reduction of the solar insolation. The solar constant $S_0$ was modified as $0.01 S_0$ in the center and assuming a linear increment in

the north-south direction. This parameterization produced a sufficient accuracy for the purposes of the experiment with 5 min in time and 10% in the amplitude of the eclipse. Vogel et al. (2001) used the Karlsruhe Atmospheric Mesoscale Model, KAMM (Adrian and Fiedler, 1991) for studying the perturbation on temperature and wind driven by the eclipse of August 11, 1999 in southern Germany. In this case, the solar constant was modified using a mathematical expression referred as "obscuration function" derived from geometric relationships in terms of the solar, lunar and observer positions. Zanis et al. (2001) used

a simple photochemical box model for investigating the response of the tropospheric ozone variations during a photolytical perturbation as in the case of that European eclipse. Szałowski (2002) built a basic model of local soil and air temperature changes in Poland for the same episode. The solar obscuration was evaluated following geometric relationships as a function of the topocentric co-ordinates of the centers of the solar and lunar disks in the equinoctial system and their angular radii of both celestial bodies. Eckermann (2007) investigated the atmospheric response to the total solar eclipse of December 4,

2002 with the high-altitude global NWP model (NOGAPS-ALPHA). In this experiment, the obscuration of the solar disk was evaluated assuming a linear variation from the center of the eclipse to the penumbra region. Related to the WRF-ARW model, Founda and Melas (2007) parameterized the eclipse of March 29, 2006 assuming a variation of the solar constant proportional to the distance from the shadow axis considered as a point moving on Earth with a specific velocity. More recently, Wu et al. (2011) used the WRF-ARW coupled with the WRF-CHEM module for analyzing the sensitivity of the tropospheric ozone and

other chemical species as well as the effects on meteorological variables to the limb darkening effect as well as the effects





on meteorological and other chemical species during the eclipse of July 22, 2009 over China. The solar eclipse effect was added modifying the solar radiation and photolysis rates using a scaling factor as a function of the latitude, longitude, time and wavelength. The degree of obscuration was evaluated as proportional to the distance to the center of the total eclipse track provided by NASA.

The recent growth in the solar energy industry has greatly increased the interest in adding additional detail to the modeling of solar irradiance variations in NWP models for use in solar resource assessment and forecasting applications. Regarding the resource forecasting, solar eclipses are episodes that increase significantly the errors because the operational shortwave schemes implemented within NWP models neglect these astronomical events. We propose a general approach for modeling the eclipse effects within the WRF-ARW model based on the Bessel's method (e.g. Chavuenet (1871)) and the Five Millennium

Catalog of Solar Eclipses provided by NASA (Espenak and Meeus, 2008). The method is widely used in many astronomical applications related to occultations and eclipses as a particular case. This approach replaces the highly complex equations describing the orbital motions of the Sun, Moon and Earth with a simpler equation set expressed in terms of the location on the Earth's surface and the position and motion of the Moon's penumbral and umbral shadows with respect to the center of the Earth. The use of this simpler description does not lose accuracy and it becomes independent of the observer coordinates.

These variables are used to evaluate the eclipse conditions at each grid-point represented by the degree of obscuration and modifying the incoming radiation accordingly.

The study has two major components. In the first part, the algorithm implemented to model solar eclipses in the WRF-ARW is described in Sect. 2 and the results of a validation of the solar eclipse trajectories computed by the algorithm with respect to published NASA values are presented in Sect. 3. The second component of the paper presents results from tests of the

new WRF-ARW algorithm and code in simulations for four eclipse cases. The validation of the simulated global horizontal irradiance with data from the Baseline Surface Radiation Network (BSRN) (Ohmura and Gilgen, 1998) is provided in Sect. 4 and the simulated eclipse-induced temperature and wind speed perturbations are described in Sect. 5.

## 2   Implementation in the WRF-ARW model

The conceptual idea for including the solar eclipses within the WRF-ARW model is similar to the previous attempts performed

by Gross and Hense (1999) or Founda and Melas (2007), among others as we explained in the Introduction. The solar eclipse occurs when the disk of the Sun is hidden partially or totally by the Moon. Consequently, this process produces a reduction of the incoming radiation at the top of the atmosphere (TOA), $S_{in}$.

The magnitude is computed in terms of the solar constant, $S_0$ and the cosine of the solar zenith angle, $\mu_0$ as

$$S_{in} = S_0\mu_0. \tag{1}$$

In the default version of the WRF-ARW model, $S_0$ is evaluated at the module named radiation_driver at each radiative call and shared with all the shortwave parameterization routines as an input variable. Given one day of the year, this variable is





assumed as a constant at all grid-points of the domain (i.e. scalar magnitude). This number is determined using a baseline solar constant of 1370 W m$^{-2}$ modulated by an eccentricity factor determined following the methodology of Paltridge and Platt (1976) as a function of the day of the year. On the other hand, $\mu_0$ is calculated using spherical astronomy equations in terms of the date, time and the geographic coordinates of the current grid-point. Therefore, Eq. 1 has a dependence on grid-point $i, j$
and on time $t$ as

$$S_{in,tij} = S_{0,t}\mu_{0,tij}. \tag{2}$$

As it is discussed in previous publications as Founda and Melas (2007), the key point in modeling the impact of solar eclipses in the WRF-ARW radiation physics is the modification of the incoming radiation by a spatially dependent (i.e. a 2-D array) degree of obscuration D. This variable takes into account the part of the solar disk that is hidden by the Moon with
a geographical and temporal dependence due to the observer perspective and the solar and lunar motions with respect to the Earth. Thus, given one time $t$, we can rewrite Eq. 2 as

$$S_{in,tij} = S_0, t\mu_{0,tij}(1 - D_{tij}). \tag{3}$$

When the grid-point is not under eclipse conditions, $D_{tij} = 0$ and Eq. 3 becomes Eq. 2. On the contrary, when a grid-point is under the totality, $D_{tij} = 1$ and $S_{in,tij}$ becomes zero.
The eclipse trajectory is determined using the Bessel's method broadly explained in several manuals as Chavuenet (1871) and briefly presented in Appendix A for readers with some astronomical background. Although this approach dates from the $19^{th}$ century, it is still being used by many institutions such as NASA.

The Bessel's method is a general approach used to predict the place and time for observing all the celestial phenomena as occultations and eclipses. In the case of the solar eclipses, this approach projects the Sun and Moon orbit trajectories on a plane
passing through the Earth's center and being perpendicular to the axis of the Moon's shadow defined such as the "fundamental plane". On this plane, a Cartesian coordinate system in $\mathbb{R}^3$ is used, with the $X$ and $Y$ axes constructed on the fundamental plane and with the origin at the Earth's center. By construction the Z-axis is normal to the fundamental plane and parallel to the axis of the shadow. This new reference system is useful because we can define a set of variables that are only relative to the fundamental plane and invariant to the observer. These magnitudes are denoted as "besselian elements" and they are detailed
in Appendix A. As the besselian elements only depend on the fundamental plane and the astronomical ephemeride or almanac, they can be evaluated before an eclipse without considering the point of view of the observer.

There are several catalogs for eclipses and occultations providing the besselian elements based on the astronomical ephemerides. For the particular case of the solar eclipses, NASA provides two catalogs: the Five Millennium Catalog of Solar Eclipses (Espenak and Meeus, 2008) that contains all partial, annular, total and hybrid eclipses from 2000 BCE to 3000
CE and the Ten Millennium Catalog of Long Solar Eclipses (Espenak and Meeus, 2009) with a period from 4000 BCE to 6000 CE.





We store these besselian elements in a WRF file named eclipse_besselian_elements.dat that should be present in the running folder for the model. This file contains a data-base of all partial, annular, hybrid and total eclipses from 1950 to 2050 (both included) based on the Five Millennium Catalog of Solar Eclipses (Espenak and Meeus, 2008).

Following the set of equations described in Appendix A, the degree of obscuration is evaluated for each grid-point at each
radiation call. Then, the incoming radiation is modified following Eq. 3 before calling the configured solar parameterization.

## 3 Algorithm validation

In order to evaluate the degree of accuracy and reliability in the eclipse computation, the proposed algorithm has been validated with respect to the NASA's values (Espenak and Meeus, 2008). As the lunar shadow has a circular shape (Eq. 22) in which each Earth point is separated by a distance $\Delta$ from the center, the evaluation of the shadow's axis is enough for determining
the degree of accuracy of the new algorithm.

The validation includes all total, annular and hybrid episodes for the period between 1950 and 2050. Partial eclipses can not be validated because the trajectory is not well-defined on the Earth's surface (Appendix A). Moreover, there are some particular cases near the poles in which the axis of the shadow does not cross the Earth surface and hence, they are mathematically undefined. These cases are not included in the validation: 1957-04-30 (Annular, North Hemisphere), 1957-10-23 (Total, South
Hemisphere), 1967-11-02 (Total, SH), 2014-04-29 (Annular, SH), 2043-04-09 (Total, NH) and 2043-10-03 (Annular, SH).

The results show a bias less than $\pm 5 \cdot 10^{-3}$ degrees for latitude and longitude, and in many cases even lower than $\pm 1 \cdot 10^{-4}$ degrees (Fig. 1), being errors in longitude significantly higher than in latitude. In general, latitude shows positive biases while longitude tends to be underestimated, in other words, the modeled eclipse experiences a small temporal delay with respect to the NASA values. There are not relevant differences between eclipse types. These errors can be associated with small differences
on the code as well as truncation errors due to the compilation options. This degree of accuracy implies a bias less than $\pm 550$ m in the Equator and decreasing with the latitude. Thus, it is enough for the majority of mesoscale applications.

## 4 Cases of study

The proposed implementation within the WRF-ARW model is tested in four real simulations with two goals: i) evaluate the degree of improvement on the GHI outcomes required for solar energy industry applications and ii) observe the degree of
realism of the WRF-ARW model response necessary for future scientific research in the line of previous works such as Founda and Melas (2007).

As real measurements, we use data from the BSRN network (Ohmura and Gilgen, 1998) because this data-set provides radiation, surface and upper-air measurements for 58 stations around the world in many climate zones and, in some cases, covering periods longer than 20 years. Moreover, the solar radiation measurements are provided with high time resolution (1
to 3 min) which are convenient for evaluating GHI performance.





The cases of study are chosen in terms of the spatial and temporal coverage of the BSRN stations and the data availability. After a previous analysis of the data-sets, four total episodes are presented: 1994-11-03 (South America), 1999-08-11 (Europe), 2006-03-29 (North Africa) and 2009-07-22 (Eastern Asia). The location of the BSRN stations used for each episode are included in Fig. 2. These stations are: Florianopolis (FLO) in South America, Carpentas (CAR), Lindenberg (LIN) and Payerne

(PAY) in Europe, Tamanrasset (TAM) in North Africa and, Tateno (TAT) and Xianghe (XIA) in Eastern Asia.

For each episode, we create a single domain composed by 200x200 grid points (Fig. 2) and 50 vertical levels with a horizontal resolution of 27 km and a top of the model at 50 hPa. All simulations are initialized using the ERA-Interim Reanalysis at 0.7° x 0.7° (Dee and Uppala, 2011) at 18 UTC on the day before the date of the eclipse in order to minimize the impact of model spin-up. Other settings related with the model configuration are described in Appendix B because this information is not relevant

for the experiments presented here.

Although the new code has an impact on all the shortwave parameterizations, we reduce the discussion to Dudhia (Dudhia, 1989) for two reasons: i) the eclipse modifies the incoming radiation that is the same for all the schemes (Sect. 2) and ii) Dudhia is the simplest shortwave parameterization available in the model and therefore, suitable for these initial experiments.

For each case, we run two simulations, one using the default version of the WRF-ARW model (release 3.6.1) used as a

"control simulation" and one using the new implementation. In both cases, the cloud interaction within the solar scheme is disabled using the parameter icloud in the namelist.input file. There are two reasons for disabling the cloud effects. On the one hand, cloud determination is one of the most important sources of error in mesoscale models and hence, they only add noise to the discussion. On the other hand, the main goal of this study is the implementation of the Bessel's method, shifting clouds to a secondary role. Moreover, the horizontal resolution used for these experiments can not produce the desirable cloud granularity

to be compared with on-site real time-series. Nevertheless, the microphysics scheme is enabled for obtaining a more realistic response of the model. In the following sections, the baseline version of the model without the eclipse physics will be referred to as "WRF3.6.1" and the model version with the eclipse algorithm will be called "WRF–eclipse".

The set of results presented in this study are provided with a 1-min time resolution in order to capture the relevant variations during the solar eclipse. Spatially, the nearest grid-point is selected for representing each BSRN site.

**5 Results**

This section includes a discussion of the results produced by the simulations described in Sect 4. The analysis is divided in two parts. First, we will evaluate the skills of the new algorithm for reproducing the solar eclipse conditions at different geographical places and different episodes (Sect. 5.1). In the second part, we will present a discussion about the model response looking for a broadening of the conclusions described in Founda and Melas (2007) including a higher number of situations (Sect. 5.2).

Before presenting the results of the analysis, two parameters used in the discussion should be defined. The first is the "First contact time in domain" (FCTD) described as the time–stamp in which one node in the domain has an obscurity degree different than zero. From that moment, both simulations are not strictly equal because the incoming radiation in WRF–eclipse has been modified. This variation has a direct impact on the solar heating rate profile as well as on the GHI and consequently, on the





other meteorological fields through i) the Euler equations and ii) the land surface model and surface layer parameterizations (Montornès, 2015). Trivially, the FCTD is common for all sites given one episode.

In the cases of study presented in this paper, we observe that the FCTD occurs ∼1 hour earlier than the first contact of the axis of the eclipse in the domain represented in Fig. 2. The physical reason can be easily interpreted considering the velocity of the axis across the Earth and the radius of the shadow. Considering the episodes sorted chronologically (i.e. 1994-11-03, 1999-08-11, 2006-03-29 and 2009-07-22), the FCTD is observed at 11:08, 08:39, 07:41 and 00:06 UTC, respectively (Table 1).

The second term that will be useful for describing the results is the "Maximum obscuration time" (MOT), time in which the obscuration degree is maximum given one site and thus, the GHI reaches the lowest value under cloudless sky assumption.

## 5.1 Global horizontal irradiance

The study of the GHI shows a similar behavior through all the analyzed sites (Fig. 3). Before and after the eclipse, the WRF3.6.1 and the WRF–eclipse show identical outcomes, while the first one reproduces the expected cloudless daily pattern during the eclipse and the second decreases abruptly reaching the minimum value at MOT and increasing again later. The reduction of the GHI depends on the eclipse conditions at each place.

Time-series for WRF–eclipse are well synchronized with respect to the real data-sets. The MOT (Table 1) in FLO is reached at 13:01 UTC. CAR, LIN and PAY show the lowest GHI at 10:28, 10:42 and 10:31 UTC, respectively. TAT and XIA have the lowest GHI at 02:14 and 01:33 UTC. Finally, TAM experiences the MOT at 9:53 with a delay of ∼1 min with respect to the real measurements. This delay is produced because we are considering the nearest grid-point in an equatorial region.

The amplitude of the GHI reduction shows a good agreement in those sites showing clear-sky conditions on real measurements (e.g. FLO, TAM) while those sites with clouds (e.g. PAY, TAT) show a tendency to overestimate the GHI because we are not considering the effect of clouds on radiative transfer in these experiments.

The accuracy is quantified in terms of the bias and the mean absolute error (MAE) defined as

$$BIAS = \frac{1}{N} \sum_{i=1}^{N} (f_i - o_i),$$
(4)

$$MAE = \frac{1}{N} \sum_{i=1}^{N} |f_i - o_i|,$$
(5)

respectively. The validation includes only those time-stamps with an obscuration greater than zero in the WRF–eclipse. Hence, $N$ is the number of valid frames while $f$ and $o$ are the modeled and real values, respectively. These metrics are not normalized with respect to the radiation at TOA as it is usually done because this variable is not the same in WRF3.6.1 as in WRF–eclipse.

Sites under cloudless conditions show the highest improvement in MAE (Table 2). In FLO, MAE drifts from 438 Wm$^{-2}$ in WRF3.6.1 to 44 Wm$^{-2}$ in WRF–eclipse (i.e. an improvement of +90%). TAM shows similar results with a MAE decreasing



from 352 Wm$^{-2}$ to 82 Wm$^{-2}$ (i.e. +77%). Both sites show a high reduction of the bias. In FLO, from 438 to -34 Wm$^{-2}$ while in TAM from 348 to -82 Wm$^{-2}$. This high underestimation is associated to the near grid-point issue mentioned before.

In contrast, cloudy sites show a lower improvement. CAR is the best one with an enhancement of +86% reducing the MAE from 364 to 50 Wm$^{-2}$, drifting from a high positive bias of 364 Wm$^{-2}$ to a slightly negative one with -42 Wm$^{-2}$.
LIN experiences a reduction of the MAE from 480 to 170 Wm$^{-2}$ (i.e. +73%) while the bias decreases from 476 Wm$^{-2}$ to 94 Wm$^{-2}$. PAY is the worst European site with an improvement of +71%. In this case, the MAE drifts from 580 to 170 Wm$^{-2}$ with a high positive bias. Finally, TAT and XIA show the worst degree of improvement. In TAT, the MAE is reduced from 798 to 395 Wm$^{-2}$ (i.e. +50%) while in XIA, MAE shifts from 493 to 176 Wm$^{-2}$ (i.e. +64%). The bias on the Asian sites (TAT and XIA) drift from 798 to 395 Wm$^{-2}$ and from 493 to 176 Wm$^{-2}$, respectively.

## 5.2 Response of the WRF-ARW model

Shortwave schemes have a remarkable role within NWP models, more significant in cloudy situations than in cloudless ones due to the approximations assumed in the computation of the radiative transfer equation. The selection of one solar parameterization or another produce differences on the heating rate profile as well as on the surface energy balance leading to variations in the other fields due to the high non-linear relationships between the dynamics and the physics of the model (Montornès, 2015).

Including solar eclipses within the mesoscale model is conceptually the same issue. The shadow of the Moon reduces the GHI as viewed in Sect. 5.1 and the heating produced by ozone and water vapor absorption in the stratosphere and troposphere, respectively. As a consequence, the surface energy balance is modified reducing the available energy to be transformed in latent, sensible and ground heats while the diabatic term in the energy equation decreases significantly producing changes in the other fields.

The following analyses are focused on the surface variables. Particularly, the surface heat fluxes, temperature at 2 m and wind speed at 10 m. Surface fluxes and temperature are analyzed because they are the most direct response to the GHI perturbations. On the other hand, the surface wind is chosen since it provides an indirect and integrated response to the GHI perturbations because it incorporates pressure gradient changes as well as variations in turbulence (i.e. stability). The discussion is focused on the same BSRN sites analyzed in Sec. 5.1, for consistency. However, the model outcomes are not compared with real measurements because the temporal resolution of the weather variables in the BSRN stations is 3-hourly given that they report these data-sets to the SYNOP network and, consequently, they can not provide the required temporal granularity to analyze the effects of the eclipse over the real atmosphere.

The response of the surface fluxes is instantaneous or less than 1 min as it is observed in Fig. 4 and similar to that of observed during a typical sunset and sunrise. Before MOT (Table 1), sensible (SH) and latent heat (LH) experience a reduction together with an increment of the ground heat (GH). At MOT, these magnitudes reach the maximum difference with respect to WRF3.6.1. The range of these deviations varies from one site and episode to the other, being related to the degree of obscuration, the moment of the day and the year (i.e. linked to the development of the PBL), land use, soil properties and weather conditions (i.e. humidity or cloud presence, among others) as indicated by Eaton et al. (1997). The departures in SH





between WRF3.6.1 and WRF–eclipse are $\sim$200 Wm$^{-2}$ following a similar behavior to that of reported by the same study using real measurements. In FLO, CAR and LIN, SH reaches slightly negative values around the MOT. LH experiences larger departures than SH, with differences larger than $\sim$250 Wm$^{-2}$ in FLO and CAR. TAM and TAT experience LH variations lower than SH due to the local meteorological features. The GH shows a significant increment reaching near-zero values or

even slightly positive as a response of the SH decay, being highly dependent on the soil features. The largest GH deviations between WRF3.6.1 and WRF–eclipse are produced in TAM because this site is located near to the Sahara desert.

After the MOT (Table 1), surface fluxes in WRF–eclipse tend to recover similar values as in WRF3.6.1. In some sites such as CAR or XIA, SH experiences greater values than WRF3.6.1. just after recovering the solar disk. Other sites with a high dependence on the daily patterns as FLO and TAM show lower SH values.

In this discussion of the fluxes, PAY is an exception. This site experiences a time-lag in the response with respect to the MOT with positive heat fluxes deviations (Fig. 4). This pattern is produced because the grid-point used for the analyses correspond to a water body given that this site is located near to a water area (i.e. Lake Neuchâtel).

Surface fluxes are important because they provide the lower boundary condition for evaluating the vertical transport parameterized within the PBL schemes. Within the model, these physical processes are parameterized in three physical packages: the

land-surface model (LSM), PBL and surface layer scheme. The LSM approximates those processes occurring at the surface (i.e. surface energy budget, evaporation and soil processes, among others) and it returns SH, LH, terrestrial emission and short-wave reflection to the atmospheric model. The PBL scheme parameterizes the vertical transport of momentum, energy and water vapor between the lower levels of the model and the free atmosphere. Both packages interact through the surface layer parameterization. This physical scheme computes the exchange coefficients and the friction velocity required for calculating

SH and LH within the LSM. Moreover, the surface layer parameterization diagnoses the temperature at 2 m and wind speed at 10 m based on the similarity theory equations (Stull, 1988).

Therefore, temperature at 2 m and wind speed at 10 m are two interesting fields for understanding the response of the model in an eclipse episode at a first order, without considering a full analysis of the PBL that might require a fully dedicated study and distant to the purposes of this one.

Variations in the temperature at 2 m show a delay with respect to the GHI and heat fluxes (Fig. 5). This response has a similar timing and magnitude as reported in studies using real measurements such as in Fernández et al. (1993a) (between 10 and 30 min) or Founda and Melas (2007) (between 10 and 15 min). FLO shows the greatest variation with -4.4 K, 12 min after the lowest GHI and SH values (Fig. 3 and 4). CAR, LIN, TAM and TAT experience a similar behavior within them. In CAR and LIN, the temperature decreases 2.9 and 2.6 K, 5 and 3 min after the MOT (Table 1), respectively. TAM shows a reduction of

2.1 K, 6 min after the maximum obscuration while in TAT, the temperature decays 2.2 K with a delay of 4 min. Finally, PAY and XIA show the largest delays with similar variations. In the first case, the minimum value is reached 1 hour and 16 min after the MOT with an anomaly of -1.1 K. In the second case, the minimum temperature is showed with a shift of 11 min and a variation of -1.2 K.

After the eclipse, all sites tend to recover similar patterns as observed in WRF3.6.1. However, there are some significant

differences. CAR, LIN, PAY and TAM are stations that experience temperatures around 0.5 and 1 K lower than in WRF3.6.1



outcomes, being PAY the most conservative case due to the lake effect. TAT shows a positive anomaly between 10:00 and 12:00 UTC falling quickly to near-zero negative departures. XIA is the site with the highest variation in the temperature drifting from positive departures (∼+0.5 K) to negative ones (∼-1 K, even ∼-1.5 K) during the successive hours with a tendency to become more stable at the end of the day. These extreme temperature variations are a consequence of the patterns described in the

surface fluxes. Finally, FLO experiences positive differences at the local midday drifting to near-zero departures at the local evening. The reason of this positive anomaly is described by the local meteorological conditions. FLO is located in the coast (Fig. 2) highly influenced by breezes producing a well-defined wind speed pattern (Fig. 6). The eclipse produces a delay in the wind speed daily maximum leading to a warmer air at the midday with respect to the simulation without the eclipse. Similar results were reported in Subrahamanyam and Anurose (2011b) comparing real measurements for an eclipse episode

with respect to a control day in a region of India highly influenced by the sea/land breezes.

The response of the wind speed varies from one site to the others (Fig. 6) linked to high non-linear relationships between the model dynamics and physical schemes.

The temporal response in FLO, CAR and LIN is similar to that of observed for the temperature (Fig. 5). These sites experience an abrupt reduction of the speed with near-zero deviations some hours after the MOT. CAR and LIN experience a

minimum wind speed 5 and 3 min after the MOT, with -1.8 and -1.6 ms$^{-1}$, respectively. Both sites show a wind speed above to WRF3.6.1 (less than +0.5 ms$^{-1}$) after the eclipse linking with the pattern observed in SH (Fig. 4). FLO shows two minima, the first is produced 3 min after the MOT and the second minimum ∼4 hours later, after the local midday. Both cases show a wind speed reduction of ∼1.5 ms$^{-1}$ being the second one less important. This observed pattern is a direct consequence of the temperature lag that debilitates the sea breeze.

PAY and TAM show near-zero positive variations at MOT with a positive peak of +0.5 and +0.8 ms$^{-1}$, 40 and 43 min after the maximum obscuration (Table 1), respectively. After this peak, wind speed shows a negative minimum, more important in PAY than in TAM. During the next hours, PAY experiences slightly lower speeds than in WRF3.6.1, while TAM maintains negative deviations. The reason of this pattern in TAM is the Saharan desert. As a consequence of the eclipse, the surface of the desert becomes cooler than in WRF3.6.1 producing a weakening of the temperature gradients and thus, experiencing lower

wind speeds during the afternoon and evening.

TAT and XIA show the highest departures between WRF–eclipse and WRF3.6.1 compared with the other stations. At the MOT, both sites experience a near-zero negative deviations. After the eclipse, the wind differences show a set of positive and negative peaks in time, noisier and larger in XIA (from ∼-3 to ∼2.5 ms$^{-1}$) than in TAT (from ∼-1 to ∼1.5 ms$^{-1}$). TAT shows two well-defined patterns (Fig. 6), one with negative departures before the sunset at ∼10:00 UTC (Fig. 3) and the other at night

with positive departures drifting to near-zero negative values at the end of the day. The reason of this pattern is again in the sea breeze and the temperature gradient. TAT is located in the coast of Japan (Fig. 2). In the simulation considering the eclipse, the land reaches lower temperatures decreasing the gradient with respect to the sea. Consequently, wind speed reaches lower values at the daytime hours than in WRF3.6.1. At night, the pattern is reversed when land surface temperatures in WRF–eclipse reach lower values faster than in WRF3.6.1, strengthening the gradient with respect to the sea and thus, producing an increment of

the wind.





## 6 Conclusions

This paper describes the implementation of a new package within the WRF-ARW model that includes the effect of the solar eclipses. The presented approach uses the Bessel's method and the Five Millennium Catalog of Solar Eclipses provided by NASA for determining the eclipse conditions at each grid-point of the domain (Appendix A). Once the position of the Sun

and the Moon with respect to the observer (i.e. position within the model domain) are computed, the degree of obscuration is evaluated following geometric relationships. This magnitude is then used for correcting the incoming radiation at each grid-point accordingly.

The new algorithm has been validated with respect to the NASA's values for the eclipse trajectory covering all the total, annular and hybrid eclipses from 1950 to 2050. This validation show a good accuracy in the determination of the latitude and

longitude for the main requirements of mesoscale model applications. Both variables are computed with a bias lower than $\pm 5 \cdot 10^{-3}$ degrees (i.e. ~550 m at the Equator) with a tendency to overestimate the latitude and underestimate the longitude.

In order to check this new implementation, the code has been tested in real simulations and compared with the default version 3.6.1. The analysis includes four total solar eclipse episodes: 1994-11-03 (South America), 1999-08-11 (Europe), 2006-03-29 (North Africa) and 2009-07-22 (Eastern Asia).

The variations in the solar radiative transfer due to the eclipse produce two impacts on the model: one in the GHI and the other on the heating rate profile. The first one has a contribution in the surface energy budget parameterized in the LSM, while the second one modifies the diabatic term in the energy equation and thus in Euler equations.

The new GHI shows a good agreement with respect to the real measurements. The modeled solar eclipse is well synchronized with the reality in all sites. The modeled eclipse-induced GHI perturbations agree very well with the measured perturbations at

sites (e.g. TAM, FLO) with observed cloudless skies but the agreement is not as good in observed cloudy scenarios (e.g. TAT, PAY) because the effects of clouds are not included in these experiments.

The reduction of the GHI leads to instant changes in the SH, LH and GH fluxes. The response of these fields varies as a function of the degree of obscuration and the land use. As a consequence, the PBL experiences changes that are represented by the temperature at 2 m and wind speed at 10 m. In general, both fields experience an abrupt reduction while the solar disk

is hidden by the Moon, faster in temperature than in wind speed. After the eclipse, all the analyzed sites tend to recover values similar to those in WRF3.6.1.

The response on the temperature at 2 m varies from ~-1 to ~-3 K with a time-lag between ~5 and ~15 min after the maximum obscuration (Table 1). In places over water bodies this delay is larger requiring more than 1 hour due to the thermal inertia of water. On the other hand, the response of wind speed at 10 m is strongly influenced on the temperature. Thus, the

solar eclipse has a heavy impact on those sites near to the coast due to the effects on the solar breeze.

The presented algorithm opens the door into a further systematic analyses regarding the gravity waves produced by the eclipse shadowm the effects on the vertical motions influenced by changes in the heating rate profile in the stratosphere or the local scale patterns by increasing the model resolution, among others.





*Acknowledgements.* The ECMWF ERA-Interim data used in this study have been obtained from the ECMWF data server.

Eclipse Predictions by Fred Espenak, NASA's GSFC.

BSRN investigators and site scientist for maintaining and improving this data network for all the scientific community.

Mrs. Imma Torras for her collaboration preparing the besselian elements for running within the WRF-ARW model.

**Appendix A**

Physically, a solar eclipse occurs when the lunar and solar centers are distant from one another by an arc in the celestial sphere equal to the sum of their radii (Buchanan, 1904). For many centuries, earlier astronomers tried to determine the occurrence of a solar eclipse based on the movement of both celestial bodies in the sky. Although they made high accuracy predictions, the method was not efficient because of the tedious computations being only valid for a given place. In the $18^{th}$ century, after the

Kepler's laws and the Principia of Sir Isaac Newton, astronomers as Edmund Halley made different eclipse predictions based on the Earth and Moon orbits but the mathematical treatment was highly complex due to the Earth and Moon's respective movements (i.e. translation, rotation, nutation) around the Sun.

At the beginning of the $19^{th}$ century, Friedrich Wilhelm Bessel developed a new method providing a high mathematical simplification and being independent on the observer. In fact, the approach is more general and it is valid to predict the place

and time for observing all the celestial phenomena as occultations and eclipses. This method is still being used in the computer algorithms used for solar eclipse predictions (e.g. NASA).

The method has been widely detailed in books and manuals such as Chavuenet (1871). However, we include a brief description of the Bessel's method in this appendix in order to contextualize the implementation within the WRF-ARW model described in Sect. 2.

The main idea of this approach is to reduce the problem to a single plane passing through the Earth's center and being perpendicular to the axis of the Moon's shadow. This plane is named "fundamental plane".

Before following with the description, let us define a Cartesian coordinate system in $\mathbb{R}^3$ in which X and Y axes are constructed on the fundamental plane, with the origin in the Earth's center. In this new reference system, let us assume the positive X-axis in the east direction and the Y-axis in the north direction. By construction, the Z-axis is normal to the fundamental plane

and parallel to the axis of the shadow. This new system is more useful than the one located at the Earth's surface because we can define a set of magnitudes relative to the fundamental plane that are independent to the observer.

These variables are the coordinates $x$ and $y$ of the point where the shadow axis crosses the fundamental plane, the direction of the shadow axis in the celestial sphere described by the declination $d$ and the ephemeride solar angle $\mu$, the radii of the penumbral and umbral shadows $l_1$ and $l_2$ in the fundamental plane and the angle that the penumbral $\alpha_1$ and umbral $\alpha_2$ shadow

cones make with the shadow axis defined by $f_1 = \tan\alpha_1$ and $f_2 = \tan\alpha_2$. This set of variables are named "besselian elements" and they only depend on time in the XYZ system. Therefore, the besselian elements can be computed before an eclipse and used to determine the episode features as the trajectory of the shadow or the visibility at any place around the Earth.





For the particular case of the solar eclipses, NASA supplies two catalogs: the Five Millennium Catalog of Solar Eclipses (Espenak and Meeus, 2008) that contains all episodes since 2000 BCE to 3000 CE and the Ten Millennium Catalog of Long Solar Eclipses (Espenak and Meeus, 2009) with a period from 4000 BCE to 6000 CE.

In these catalogs, NASA provides for each eclipse a reference time $t_0$ in the Terrestrial Dynamical Time (TDT) reference system and a set of polynomial coefficients to compute the besselian elements valid in a 6-h period centered in $t_0$ (i.e. $t_0 \pm 3$).

For a given eclipse and time $t_1$ in TDT, the besselian elements are evaluated as

$$x = x_0 + x_1 t + x_2 t^2 + x_3 t^3, \tag{6}$$

$$y = y_0 + y_1 t + y_2 t^2 + y_3 t^3, \tag{7}$$

$$d = d_0 + d_1 t + d_2 t^2, \tag{8}$$

$$\mu = \mu_0 + \mu_1 t + \mu_2 t^2, \tag{9}$$

$$l_i = l_{i,0} + l_{i,1} t + l_{i,2} t^2, \tag{10}$$

with $i=1,2$ (penumbra and umbra) and where $t = t_1 - t_0$ in TDT. Note that the cone angles $f_1$ and $f_2$ are assumed constant during all the eclipse (i.e. $t_0 \pm 3$ h). By construction, the penumbra shadow radius in the fundamental plane $l_1$ is always defined as a positive value, while the umbra shadow radius $l_2$ is defined as positive for annularity and negative for totality.

Internally, the WRF-ARW model considers the time in the Coordinated Universal Time (UTC) system. Therefore, before computing the besselian elements, this time should be converted to TDT. This conversion is performed using a variable that astronomers call "delta-T" or, hereinafter, $\Delta t$. Conceptually, this parameter is a correction on time due to the differences on the Earth rotation produced by the angular momentum transferred from Earth to the Moon by the tidal friction. This variable is also provided in the catalogs for eclipses.

Thus, $t$ is computed as

$$t = t_1^{TDT} - t_0^{TDT} = (t_1^{UTC} - \Delta t) - t_0^{TDT}. \tag{11}$$

The key point in order to implement the eclipses in a NWP model is the determination of the degree of obscuration $D$ of the solar disk at each grid-point of the domain (Sect. 2). Each grid-point is characterized by two geographical coordinates given by the latitude $\phi$ and longitude $\lambda$. Therefore, first of all, we need to transform this pair of coordinates into the reference system XYZ.

Nevertheless, we need to introduce a couple of corrections to the geographical coordinates provided by the atmospheric model. As the real Earth is an ellipsoid, we need to correct the geographical latitude with the eccentricity, $\epsilon$ as

$$\tan \phi_1 = \tan \phi \sqrt{1 - \epsilon^2}, \tag{12}$$

where $\epsilon$ is taken as 0.0818192 from Meeus (1991).


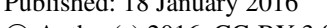


On the other hand, the geographic longitude referred to the Greenwich meridian $\lambda$ must be transformed to the ephemeride longitude $\lambda_1$ by applying the correction

$$\lambda_1 = \lambda + 1.002738 \frac{15\Delta t}{3600}. \tag{13}$$

Then, if $\xi$, $\eta$ and $\zeta$ are the coordinates of the observer in the XYZ reference system, we can express the coordinate transform as

$$\xi = \cos\phi_1 \sin H, \tag{14}$$

$$\eta_1 = \frac{\eta}{\rho_1} = \sin\phi_1 \cos d_1 - \cos\phi_1 \sin d_1 \cos H, \tag{15}$$

$$\zeta_1 = \frac{\zeta}{\rho_2} = \sin\phi_1 \cos d_2 - \cos\phi_1 \sin d_2 \cos H, \tag{16}$$

where $H$ is the hour angle in the observation place (i.e. grid-point) defined as

$$H = \mu - \lambda_1 \tag{17}$$

and $\rho_1$, $d_1$, $\rho_2$ are $d_2$ are a set of variables given by the following relationships

$$\rho_1 \sin d_1 = \sin d, \tag{18}$$

$$\rho_1 \cos d_1 = \cos d \sqrt{1 - \epsilon^2}, \tag{19}$$

$$\rho_2 \sin d_2 = \sin d \sqrt{1 - \epsilon^2}, \tag{20}$$

$$\rho_2 \cos d_2 = \cos d. \tag{21}$$

In the fundamental plane, the eclipse conditions of a grid-point ($\xi$,$\eta$) are determined by the distance $\Delta$ to the shadow axis ($x$,$y$). Thus

$$\Delta^2 = (x - \xi)^2 + (y_1 - \eta_1)^2. \tag{22}$$

Eq. 22 defines a circle centered on the shadow axis and concentric to the circles defined by the penumbra and umbra radii, $l_1$ and $l_2$. Here $y_1$ is a correction on $y$ evaluated as

$$y_1 = \frac{y}{\rho_1}. \tag{23}$$




Typically, the observer will be in a plane parallel to the fundamental plane (i.e. $\zeta \neq 0$), named as the "observer's plane". As the shadow produced by the Moon is a cone, we need to project the penumbra and umbra radii from the fundamental to the observer's plane (Fig. 7). Based on trigonometric relationships, we can demonstrate that the penumbra $L_1$ and umbra $L_2$ radii in the observer's plane are given by

$$L_i = l_i - \zeta_1 f_i, \tag{24}$$

with $i = 1, 2$.

Therefore, from Eqs. 22 and 24, we can define three regions determining the eclipse conditions at the observer's plane. First, when

$$L_1 < \Delta, \tag{25}$$

the grid-point is located out of the shadow and hence, the eclipse is not observable. Second, when

$$L_1 \geq \Delta > |L_2|, \tag{26}$$

the observer is within the penumbra region. And finally, if

$$|L_2| \geq \Delta \geq 0, \tag{27}$$

then the node is inside the umbra region.

Therefore, from these ideas along with geometric relationships, we can determine the degree of obscuration $D$ of the solar disk. Formally, we define $D$ as the part of the solar disk that is hidden by the Moon. Let us assume an observer located at a point Q inside the penumbra region with a distance $\Delta$ with respect to the axis of the shadow (Fig. 7). In a situation without eclipse, the observer $Q$ measures the total length of the solar disk as $AC$. However, during an eclipse, the lunar disk intercepts some of the solar beams and consequently, a part, $AB$, of the solar disk is not visible from $Q$. Then, the $D$ can be defined mathematically as the ratio between distances $AB$ and $AC$ as

$$D = \frac{AB}{AC}. \tag{28}$$

Note that if the observer moves outward to the penumbra region, the length $AB$ will be shorter until reaching a point in which $\Delta$ becomes $L_1$ and the distance AB is zero. At this point, solar and lunar limbs are in contact but the solar disk is not hidden (i.e. $D=0$). On the other hand, when $\Delta$ becomes less than $|L_2|$, the solar disk is completely hidden by the Moon (i.e.



solar beams can not reach the Earth's surface). In this case, the observer is inside the umbra region and it will experience the annularity or totality depending on the sign of $L_2$.

This description can be quantified expressing $D$ as

$$D = \frac{QL_1}{L_1 L_2}. \tag{29}$$

5    This equation can be approximated as

$$D = \frac{L_1 - \Delta}{L_1 + L_2}. \tag{30}$$

Note that by construction, annular solar eclipses always have a denominator greater than the numerator. Therefore, $D$ is always lower than the unity. In contrast, total solar eclipses reach the unity when $\Delta = |L_2|$ because $L_2$ is a negative value.

In the validation of the algorithm discussed in Sect. 3, the eclipse trajectories are evaluated with respect to the NASA's
10  values. Thus, we need to determine the geographic coordinates of the axis of the shadow over the Earth's surface.

By construction, all points with $\Delta = 0$ are in the axis of the shadow, or in other words, all points with $\xi = x$ and $\eta_1 = y_1$ are in the eclipse trajectory.

Therefore, the problem is reduced to find the pairs of geographical coordinates $\phi$ and $\lambda$ for each $x$ and $y_1$. Mathematically, we can write the following equation system

$$\sin\beta \sin\gamma = x, \tag{31}$$

$$\sin\beta \cos\gamma = y_1, \tag{32}$$

$$c\sin C = y_1, \tag{33}$$

$$c\cos C = \cos\beta, \tag{34}$$

$$\cos\phi_1 \sin H = x, \tag{35}$$

$$\cos\phi_1 \cos H = c\cos(C + d_1), \tag{36}$$

$$\sin\phi_1 = c\sin(C + d_1), \tag{37}$$

$$\tan\phi = \frac{\tan\phi_1}{\sqrt{1 - \epsilon^2}}, \tag{38}$$

$$\lambda_1 = \mu - H, \tag{39}$$

$$\lambda = \lambda_1 - 1.002738\frac{15\Delta t}{3600}. \tag{40}$$

25  This system of equations has a solution when $|\sin\beta| < 0$. This occurs when the shadow axis passes through the Earth's surface (i.e. total, annular and hybrid eclipses). In the partial eclipses and some total and annular polar eclipses $|\sin\beta| > 0$ and the trajectory is not defined over the Earth's surface.



## Appendix B

In this Appendix, we detail the model configuration used for the experiments as an extension of the description presented in Sect. 4. All domains are composed by 200x200 points with a resolution of 27 km and 50 vertical levels automatically distributed by the model. The top of the model is set at 50 hPa.

5    The projection used at each domain depends on the BSRN location. In the eclipses of 1994-11-03 (South America), 2006-03-29 (Africa) and 2009-07-22 (Asia), we set a Mercator geographical projection while for the eclipse of 1999-08-11 (Europe), we used a Lambert Conic Conformal geographical projection tangent to the standard latitude.

All simulations use the same physical schemes. For microphysics the WRF Single–moment 5–class Schemes (Hong et al., 2004) is used. Radiative processes are parameterized using the RRTMG Iacono and Delamere (2008) for the terrestrial part
10   of the spectrum and Dudhia (Dudhia, 1989) for the solar part as indicated in Sect. 4. Radiative transfer codes are called every minute. Surface processes are modeled with the Unified Noah Land Surface Model (Tewari et al., 2004). The vertical transport is parameterized in terms of the Yonsei University Scheme for the PBL based on Hong et al. (2006), called at every time-step. The interaction between the LSM and the PBL is performed by the MM5 Similarity Scheme (Paulson, 1970; Dyer and Hicks, 1970; Webb, 1970; Beljaars, 1995; Zhang and Anthes, 1982). As we set a coarse horizontal resolution, the Kain–Fritsch scheme
15   (Kain, 2004) option for cumulus is also enabled.

Regarding the dynamics, default settings are used in all the experiments.





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

| Episode | FCTD (UTC) | Site | MOT (UTC) |
|---------|-----------|------|-----------|
| 1994-11-03 | 11:08 | FLO | 13:01 |
| 1999-08-11 | 08:39 | CAR | 10:28 |
| | | LIN | 10:42 |
| | | PAY | 10:31 |
| 2006-03-29 | 07:41 | TAM | 09:31 |
| 2009-07-22 | 00:06 | TAT | 02:14 |
| | | XIA | 01:33 |

**Table 1.** Overview of the results. FCTD is the "First contact time in domain" while MOT is the "Maximum obscuration time".





| Episode | Site | Duration [min] | WRF3.6.1 Bias [Wm$^{-2}$] | WRF3.6.1 MAE [Wm$^{-2}$] | WRF–Eclipse Bias [Wm$^{-2}$] | WRF–Eclipse MAE [Wm$^{-2}$] |
|---|---|---|---|---|---|---|
| 1994-11-03 | FLO | 160 | 438 | 438 | -34 | 44 |
| 1999-08-11 | CAR | 167 | 364 | 364 | -42 | 50 |
| | LIN | 159 | 478 | 480 | 94 | 130 |
| | PAY | 166 | 580 | 580 | 164 | 170 |
| 2006-03-29 | TAM | 153 | 348 | 352 | -82 | 82 |
| 2009-07-22 | TAT | 153 | 798 | 798 | 395 | 395 |
| | XIA | 140 | 493 | 493 | 176 | 176 |

**Table 2.** Improvement of the GHI.





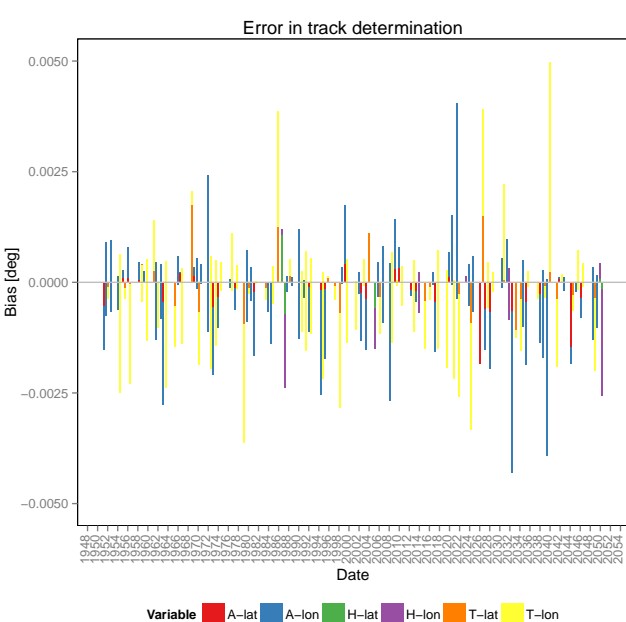

**Figure 1.** Bias in the eclipse's track computed by WRF-ARW in comparison to the NASA values. A, H and T mean annular, hybrid and total, respectively. Labels lat and lon mean latitude and longitude



**Figure 2.** Domains used for each case study. Sites are indicated in orange. The eclipse track is plotted in gray and red in order to indicate some eclipse times (UTC).





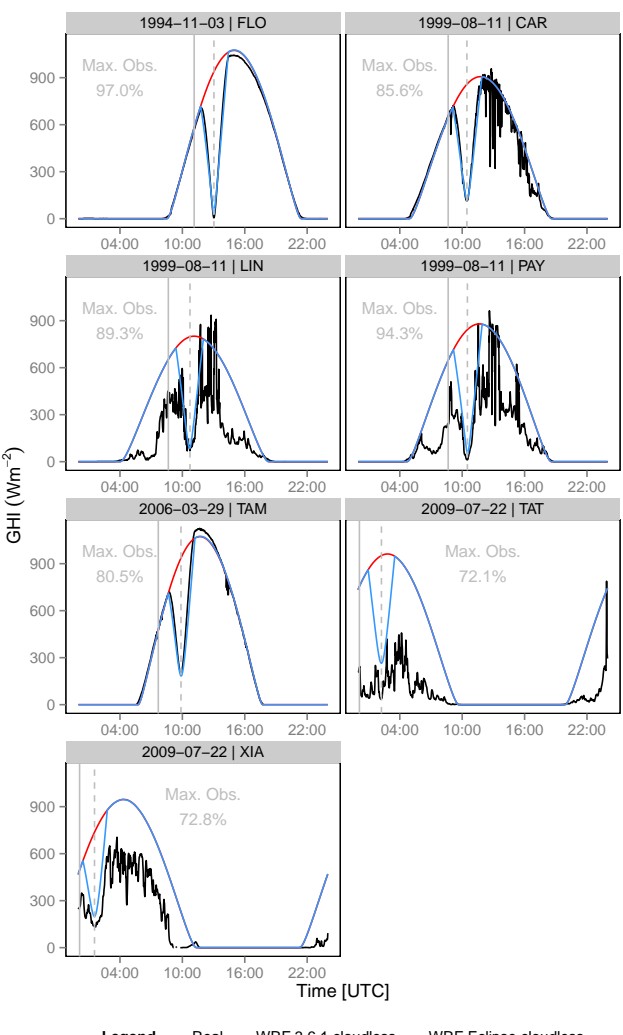

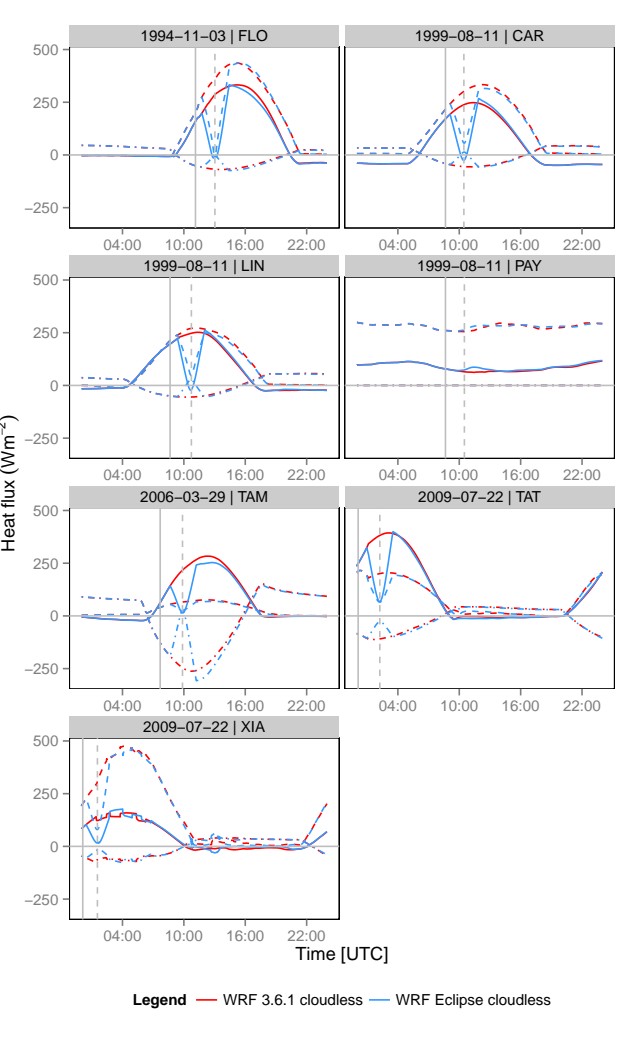

**Figure 3.** GHI outcomes for episode and site. Each total solar eclipse event is labeled with the date. Plots show real measurements (in black), the control simulation (red) and the new implementation (blue). The vertical solid and dashed gray lines indicate the time of FCTD and maximum obscuration, respectively. All results are expressed with 1-min time resolution.

**Figure 4.** Sensible heat (SH), latent heat (LH) and ground heat (GH) flux outcomes for episode and site. Each total solar eclipse event is labeled with the date. Colors indicate the control simulation outcomes (red) and the new implementation (blue). The vertical solid and dashed gray lines indicate the time of FCTD and maximum obscuration, respectively. All results are expressed with 1-min time resolution.



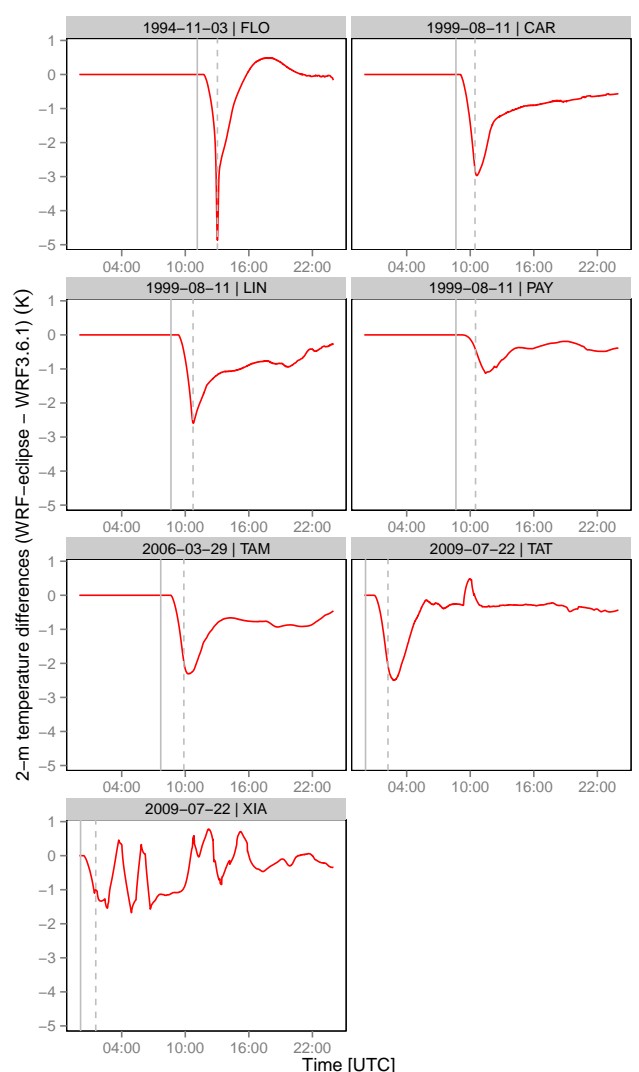

**Figure 5.** Temperature differences at 2 m for episode and site. Each total solar eclipse event is labeled with the date. The vertical solid and dashed gray lines indicate the time of FCTD and maximum obscuration, respectively. All results are expressed with 1-min time resolution.

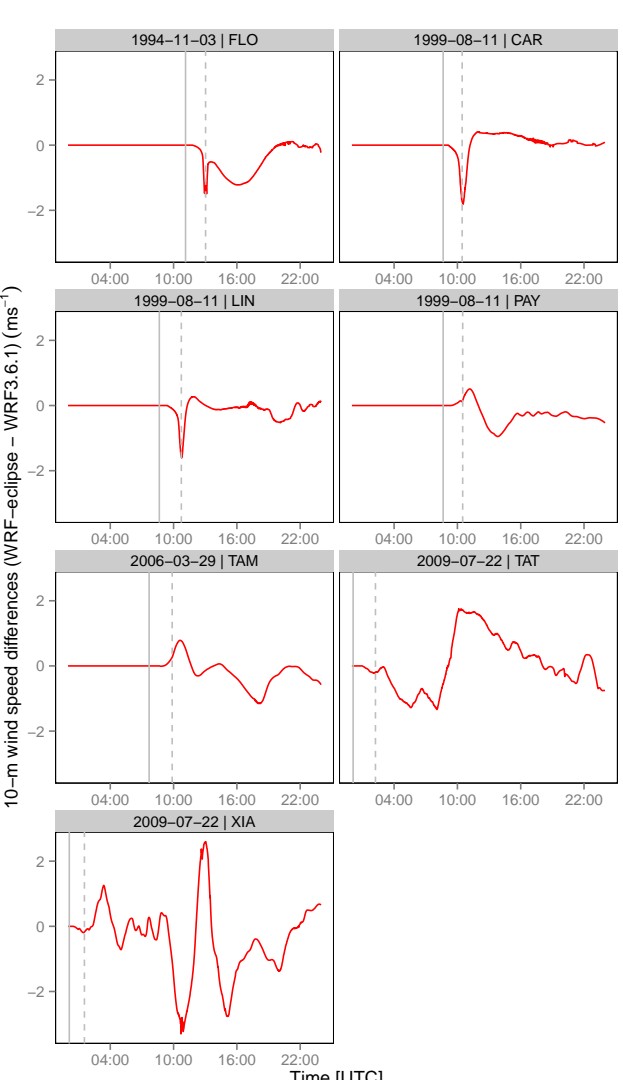

**Figure 6.** Wind speed differences at 10 m for episode and site. The vertical solid and dashed gray lines indicate the time of FCTD and maximum obscuration, respectively. All results are expressed with 1-min time resolution.



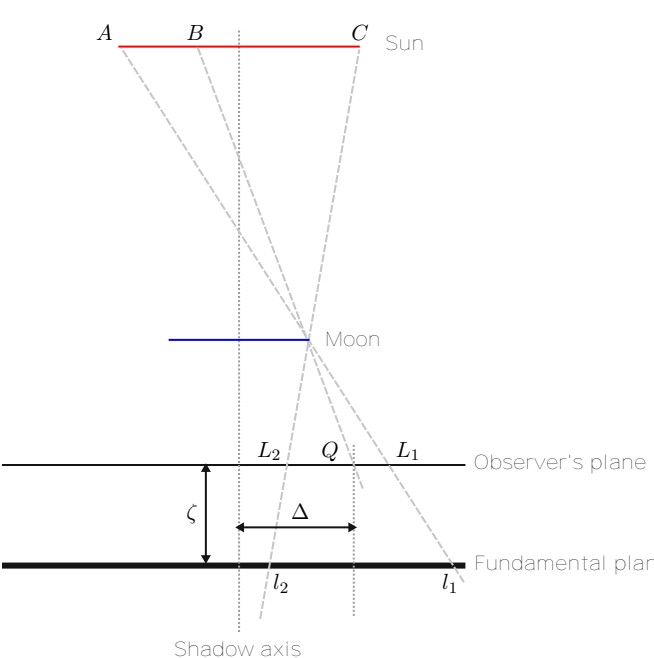

**Figure 7.** Scheme showing the geometric relationships for determining the degree of obscuration, $D$, in a total solar eclipse. A similar scheme can be drawn in an annular eclipse