# Peer review of "Implementation of the Bessel's method for solar eclipses prediction in the WRF-ARW model"

_Atmospheric Chemistry and Physics, 2015_

## Referee Comment (RC1) · Anonymous Referee #1 · 8 Feb 2016

General comments

Astronomical phenomena like solar eclipses provide a unique opportunity for the study of the atmosphere and its response under such abrupt events. From this point of view, the subject of this research is interesting, as it incorporates the eclipse events in a mesoscale model. The authors use the Bessel's method for first time in the WRF-ARW model and evaluate the model's performance. The advantages and deficits from the use of Besselians elements must be further highlighted and compared to other methods. Moreover, the performance of the eclipse-WRF as regards the response of surface layer response (surface air temperature and wind speed) must be further analysed and discusses, since no comparison with real measurements is performed. Results should be compared against findings from other studies.
[Figure]

Specific comments

Abstract should be more informative as regards the main findings and conclusions of the study. It should contain at least one quantitative information, as for instance information on the validation of the Bessels's method or GHI improvement.

A schematic representation of besselians elements showing the outline of umbra during eclipse on Earth's surface and its projection on the fundamental plane would be helpful.

Apart from wind speed, wind direction has also been proven to undergo changes in many events of solar eclipses. The authors should mention on wind direction sensitivity as well (if any), from the simulations of eclipse and control WRF.

Tables 1 and 2 should be more informative (their titles as well). The first contact, MOT and last contact should be all included in tables. Total cloud cover at the stations during the eclipse must be provided as well.

Instead of FCTD and MOT, it would be useful to see in Figures an additional vertical line corresponding to the last contact (the time after which, obscuration percentage becomes zero again). Changes in meteorological variables after the end of the episode and relevant time lags and delays should also be discussed, commented and compared with other relevant studies.

It would be helpful, to include a table in which you can illustrate together all results of WRF-ARW response. For instance, maximum changes in air temperature or wind speed, time of occurrence, time lag etc. for all stations.

In the discussion of the results, the authors should also evaluate and comment their findings against the results from other relevant studies.

The response of the model is estimated only from the differences between the 'eclipse' and control simulations. However, the model performance can't be evaluated from real measurements, because the temporal resolution of the weather variables (air temperature and wind) in the BSRN stations is 3-hourly and thus they do not provide the required temporal resolution for an eclipse event. Given the density of climatic stations (especially in Europe), I was wondering on the availability of higher resolution meteorological data from neighbor (highly correlated) stations. A discussion about the daily (24-hour) variation of air temperature and wind speed/direction from SYNOP reports at BSRN stations and detection of possible differentiations between days before/during/after the eclipse should be of some value as well as regards the real response of surface layer at the sites of interest.

Some syntax and technical issues

Line 5 (page 1): avoid the phrase '..adding additional..'

Page 2, lines 2-4 , rephrase

Page 3 , line 16. Leave space Page 5, line 12, , equation (3): t must be subscript, please correct

References of the same author should be put in chronological order (e.g Fernandez et al 1993a,b 1996).

Some references are wrong or incomplete, e.g.: Founda and Melas (2007) should be cited as Founda et al. (2007) in the text and as following in the reference list : Founda, D., Melas, D., Lykoudis, S., Lisaridis, I., Gerasopoulos, E.,Kouvarakis, G., Petrakis, M., and Zerefos, C.: The effect of the total solar eclipse of 29 March 2006 on meteorological variables in Greece, Atmos. Chem. Phys., 7, 5543-5553, doi:10.5194/acp- 7-5543-2007, 2007.

Also, Montorne (2015) and Fernandez 1993b seem to be wrong or incomplete in the text and reference list.

Figure 1, x-axis : labels appearance should be improved

Page 9 lines..1-10. The presentation of these results are somewhat confusing. Phrases

like..'the worst europena site..' must be avoided, please rephrase.

Fig 4: GH is not included in station PAY?
[Figure]

---

## Referee Comment (RC2) · Anonymous Referee #2 · 15 Feb 2016

General comment

Adding solar eclipse parameterizations to NWP models is not new, however, as pointed out by the authors, previous approaches were lacking generality and were usually designed for a particular case, only. The authors use the Bessel's method for the first time in a state-of-the-art NWP model and evaluate its performance. This work is interesting in the light of a growing demand for operational solar radiation variation forecasts (by the solar energy industry), which requires a general approach like the one described here. The paper is well structured and written. The model validation, however, should be improved. No comparison with real measurements regarding the surface layer response is included and only an idealized model setup was used (no cloud-radiation interaction and very coarse horizontal resolution). The paper can be considered for

publication after the following points have been addressed:

Specific comments

Abstract

- apart from providing a basic introduction into the topic, the abstract should focus on results/main findings. The latter is essentially lacking in the abstract. My suggestion would be to replace or extend the last section of the abstract (line 16 - 21, page 1) by the main findings. Currently lines 16-21 are basically a repetition of the introductory lines 17-22 of page 4.

Implementation in the WRF-ARW model

- The advantage of using besselian elements as compared to previous approaches (e.g. Founda et al.) should be described in more detail. Is the new approach more accurate? In what sense is it more general – or – in what sense do former approaches require manual changes to the code or input data, restricting them to single selected eclipse events?

- please elaborate a bit more on technical details of the implementation: what is the overhead/cost for this parameterization, including the setup phase (file read) in percentage of wall clock time? What is the size of the file containing the besselian elements which is read by the model? Is it read once, or opened and (partly) read at each radiation call?

Algorithm validation

The authors mention, that there are some differences between the eclipse tracks computed within the WRF module and the NASA values. However, only very vague explanations are given like "associated with small differences on the code" and "truncation errors due to compiling options". Even though the differences are relatively small, they seem to be beyond simple truncation errors. To convince the reader of the correctness of the implementation, please point out the reasons for the observed differences more clearly. Do the differences decrease significantly when performing the computations in double precision as compared to single precision?

Results

Given the fact that even global models today use resolutions of $\mathcal{O}10\,\mathrm{km}$ operationally and even higher resolutions in less time-critical scientific applications, the resolution chosen here seems rather coarse. One consequence of this coarse resolution is that the cloud interaction has been switched off, since reproducing the observed cloud structure is impossible anyway at this resolution. This approximation may be OK for qualitative comparisons against the measured GHI values and for evaluating the qualitative response of the WRF model as it is done here. The model setup chosen for validation is, however, very different from what would be used in real applications.

The paper would benefit a lot from at least one fully fledged high-resolution run (i.e. with cloud-radiation interaction switched on) and comparison of the surface layer response against real measurements (surface temperature, wind, . . . ). One may e.g. choose the europe episode, run the model for a shrinked model domain over central europe and do a validation for Lindenberg for which high resolution data should be available.

Technical issues

- Typo in Reference to Chauvenet et al. : Chavuenet $\longrightarrow$ Chauvenet

- Fig. 1: looks a bit crowded due to the use of $6$ different colors. It may improve when using only 3 different colors for A/H/T and 2 different line styles for lat/lon.

- Fig. 2: Please add some additional shading, indicating e.g. the totality zone or $90\,\%$ obscuration area. This may help the unexperienced reader to assess more easily to what degree the various stations are affected by the eclipse.

- Table 1: please add the maximum degree of obscuration for each site.

- page 9, line 29: superfluous "of"

- page 11, line 27: superfluous "a" at "a near-zero . . . "

- page 12, line 5 "the observer (i.e. position within the model domain)". Please add the description given brackets already at some previous occurrrence of "observer" in order to better clarify what is meant by "observer".

- page 12, line 9: "This validation show . . . " missing "s"

- page 12, line 32: typo "shadowm"

Model description:

Please add some information about

- the source of the applied boundary data and its update frequency (so far only the initialization is described)

- the time step

- since a lake-point was selected for comparison with the station PAY, please document whether a lake model/lake parameterization was used by WRF (i.e. is the lake (surface) temperature prognostic, or is it constant (like SST)?

---

## Author Comment (AC1) · 10 Mar 2016

**Response to "Reviewer Comments", Anonymous Referee #1**

We would like to appreciate all ideas and proposals given by Anonymous Referee #1 in order to improve the research discussed in the manuscript. In this document, we will answer each idea and we will discuss each suggestion from our point of view.

We have decided to give a personalized reply to each referee. However, some points are common for both or they have a full impact on the entire paper. For this reason, we will present firstly a common block and then we will discuss each review point by point. Hereinafter, we will use R#1 and R#2 as Referee #1 and #2, respectively.

In order to contextualize the response, the referee's commentary appears before our answer. Each review is quoted in gray. Our response appears with A: (from Authors) at the beginning and in black color. Each one is identified with a label composed by a number and a tag: GC (General Comments) and SC (Specific Comments). For example, SC4 refers to the 4th specific comment. During the discussion, the reader can find some cross-references between responses for R#1 and R#2. For example, **SC7 R#1** means the 7th specific comment of R#1.

Some answers that are also addressed to the Editor when we defend our position but we also think that the position of the reviewer is interesting. These responses are: **SC2 R#1**, **SC7 R#1**, **SC4 R#2**.

Regarding the submission of the revised manuscript, we wait until the final Editor's decision. At that moment, we will finish the last modifications and updates and we will submit the new version.

Common comments

**A:** The main idea behind the study presented in this manuscript was to discuss a new package for the WRF model (extensible to other General Circulation Models in the future) capable of representing any solar eclipse for any configuration in terms of date, domain size, grid resolution or projection, among others.

Under this framework, the manuscript was divided in three parts. The first part describes the implementation of the Bessel's method in the WRF model (Sect. 2) and it includes a validation of the algorithm by comparing our eclipse trajectories and the NASA's data-set (Sect. 3). The second part includes a validation of four real study cases for the GHI that is the most directly affected variable. The selection of these episodes was based on the availability of BSRN stations. This analysis has two major goals: i) to provide a complementary validation of the algorithm for demonstrating that

the degree of obscuration is well determined and it correlates in time with the real data and ii) to show the potentiality of this method for solar energy forecasting applications (Sec. 5.1). Finally, the last part of the manuscript introduces a brief description of the model response to show the applicability of this package for future academia research (Sec. 5.2).

The idea of including solar eclipses in the model born from the last partial and total solar eclipses that occurred in USA (October 23, 2014) and Europe (March 20, 2015), respectively, showing the necessity to incorporate these events into the solar parameterizations for solar renewable energy industry.

The main scope of this manuscript is the validation of the eclipse implementation by means of the trajectory and the GHI. From our understanding, some of the proposals of the referees for extending the study to the meteorological fields are very valuable as well as interesting but they are more appropriated as a future work. By including these extended analyses in the current version may blur the main flow of the study and may lead to a very large text.

The code used in this study has been shared with NCAR and we expect that it will be included in the next release (April, 2016). Consequently, the scientific community will be able to use this algorithm for a deep analysis of the model response and comparing with real measurements.

Further on the proposals for an extended study, both referees agree in the following aspects:

i) They suggested that it is necessary a different approach for the abstract (**SC1 R#1** and **SC1 R#2**) by including the main findings and conclusions as well as some qualitative results. We completely agree with this idea and we will rewrite the abstract. Following the idea provided by **SC1 R#2**, we have modified the last paragraph as:

"This contribution is divided in three parts. First, the implementation of the Bessel's method is validated for solar eclipses in the period 1950-2050, by comparing the shadow trajectory with values provided by NASA. Latitude and longitude are determined with a bias lower than 5 $10^{-3}$ degrees (i.e., ~550 m at Equator) being slightly overestimated and underestimated, respectively. The second part includes a validation of the simulated Global Horizontal Irradiance (GHI) for four total solar eclipses with measurements of the Baseline Surface Radiation Network (BSRN). The results show an improvement in MAE from 77% to 90% under cloudless skies. Lower agreement between modeled and measured GHI is observed under cloudy conditions since the effect of clouds is not considered in the radiative transfer schemes of the simulations. Finally, an introductory discussion of the response of meteorological variables (e.g. temperature, wind speed) to the reduction of GHI and shortwave heating rate is provided by comparing WRF-eclipse outcomes with control simulations."

ii) In **GC2 R#1** and **GC1 R#2** both referees indicated that they miss some comparison of the meteorological fields analyzed in Sect. 5.2 with real measurements. We agree with them that this kind of analysis would be interesting. Nevertheless, from our understanding, the best approach is to focus the current manuscript on the implementation of the method and prepare a future study for a better understanding of the atmosphere response with a large number of episodes and comparing with surface and vertical profile measurements.

iii) They indicated that Fig. 1 should be improved (**SC16 R#1** and **SC8 R#2**). We agree that this figure is awkward and useless with the current presentation. We have changed it as indicated in Fig. 1. We are opened to include new suggestions if necessary.

**Response to general comments**

**GC1 R#1:** Astronomical phenomena like solar eclipses provide a unique opportunity for the study of the atmosphere and its response under such abrupt events. From this point of view, the subject of this research is interesting, as it incorporates the eclipse events in a mesoscale model.

The authors use the Bessel's method for first time in the WRF-ARW model and evaluate the model's performance. The advantages and deficits from the use of Besselians elements must be further highlighted and compared to other methods.

**A:** Basically, there are two methods for computing the solar eclipses. The first one, largely used by the ancient astronomers with highly accurate results, "consists in finding the times when the disks of the Sun and the Moon are tangent in a visual line from the observer or, in other words, when the centres of the Sun and Moon are distant from another by an arc in the celestial sphere equal to the sum of their semidiameters" as it is discussed in Buchanan (1904). We have quoted the definition because it is quite explicit. On the other hand, the Bessel's method meant a simplification in the mathematical treatment and it was more useful because it is independent of the observer, therefore, it can be computed previously and applied to each place. Nowadays, all the astronomy almanacs are based on the Bessel's method. Consequently, any previous work that included solar eclipses in GCM was indirectly related to this method.

Unlike other previous works, we store the Besselian elements for 100 years and then, we evaluate the eclipse conditions at each grid-point during the simulation. The advantage of this implementation is the compatibility for any domain size, grid-resolution, projection and nest. Although the implementation of solar eclipses consumes computational resources, this deficit is negligible, as it is described in **SC3 R#2**.

A related response to this comment can be found in **SC2 R#1** and **SC2 R#2**.

[Figure]

Buchanan, R. The mathematical theory of eclipses according to Chauvenet's transformation of Bessel's method explained and illustrated, to which are appended Transits of Mercury and Venus and Occultations of fixed stars. Philadelphia and London, J. B. Lippincott company. 226 p. 1904.

**GC2 R#1:** Moreover, the performance of the eclipse-WRF as regards the response of surface layer response (surface air temperature and wind speed) must be further analysed and discusses, since no comparison with real measurements is performed. Results should be compared against findings from other studies.

**A:** As we will discuss in **SC8 R#1**, we think that this comparison is partially performed. However, we agree that it can be better developed. We will include more details in the revised version.

**Response to specific comments**

Abstract

**SC1 R#1:** Abstract should be more informative as regards the main findings and conclusions of the study. It should contain at least one quantitative information, as for instance information on the validation of the Bessels's method or GHI improvement.

**A:** We agree with this comment. As we point out in the common block, we have rewritten the last paragraph of the abstract to include some key results.

Methodology

**SC2 R#1:** A schematic representation of besselians elements showing the outline of umbra during eclipse on Earth's surface and its projection on the fundamental plane would be helpful.

**A:** We understand your point of view. However, from our perspective, we are not sure about if this kind of information is really necessary for the manuscript.

The Besselian's method is an approach existing since the 19th century. After the Newton's laws, people as Edmond Halley tried to understand solar eclipses from the law of universal attraction. Logically, the mathematical formalism for this method was extremely hard and complicated. In 1824, Friedrich Bessel proposed the method that has his name. Basically, the method projects the complicated orbits in a plane crossing the Earth's center referred as fundamental plane. The Besselian's method was a revolution in that time because simplifies enormously the mathematical treatment of the problem.

In fact, the method is used in the evaluation of transits and occultations. One of the particular applications of this method is the prediction of solar eclipses. As the method is widely used in many applications, it is fully described in many manuals. Nevertheless, we included Appendix A because we think that most of the readers of ACP/ACPD are not familiarized with astronomy. We are not really sure about if this Appendix must be included or not. But, if it is included, we think that is not necessary more details because the reader can search the information in the manuals cited in the manuscript. However, we will proceed as the Editor thinks that is the best option for the ACP journal.

Results

**SC3 R#1:** Apart from wind speed, wind direction has also been proven to undergo changes in many events of solar eclipses. The authors should mention on wind direction sensitivity as well (if any), from the simulations of eclipse and control WRF.

**A:** We agree that wind direction can improve the interest and robustness of the last part of the manuscript. Moreover, it can add more complementary value to the wind speed. We will include this information in the new version of the manuscript by adding new text and a new figure in the results.

**SC4 R#1:** Tables 1 and 2 should be more informative (their titles as well). The first contact, MOT and last contact should be all included in tables. Total cloud cover at the stations during the eclipse must be provided as well.

**A:** We agree that the captions for Table 1 and 2 can be more descriptive in order to be more useful for the reader. We will rewrite them. We also agree that they can be more informative by including other features such as the last contact.

Regarding the total cloud cover, in our understanding, this information it is not relevant for the current work. On the one hand, interaction between radiation and clouds is disabled for a better discussion of the impact of the solar eclipse in the GHI, i.e. the model does not have cloud cover affecting the radiation. On the other hand, the total cloud cover in the real data-sets may be interesting but it does not include extra information and the reader can be lost in details that are not really significant for the scope of the study.

**SC5 R#1:** Instead of FCTD and MOT, it would be useful to see in Figures an additional vertical line corresponding to the last contact (the time after which, obscuration percentage becomes zero again). Changes in meteorological variables after the end of the episode and relevant time lags and delays should also be discussed, commented and compared with other relevant studies.

**A:** We think that this information can be very interesting. We will include it in the plots. We will denote it as Last Contact Time in Domain (LCTD).

**SC6 R#1:** Changes in meteorological variables after the end of the episode and relevant time lags and delays should also be discussed, commented and compared with other relevant studies.

**A:** We agree that this part should be developed. We will incorporate this information in Sec. 5.2.

**SC7 R#1:** It would be helpful, to include a table in which you can illustrate together all results of WRF-ARW response. For instance, maximum changes in air temperature or wind speed, time of occurrence, time lag etc. for all stations.

**A:** We understand your point of view. However, we are not sure if it is really necessary because this information appears in the text as well as in figures and consequently, a new table with a summary of the results can be redundant. Nevertheless, as this is a formal aspect, we will wait until the Editor's decision if he thinks that it can be useful for the reader.

**SC8 R#1:** In the discussion of the results, the authors should also evaluate and comment their findings against the results from other relevant studies.

**A:** From our point of view, this information is already included in Sec 5.2. For example, in Pag 10, Line 25-33, we mentioned the findings of other authors. The main problem of this kind of comparison is that episodes occur at different regions of the world with very different climate features making more difficult a systematic comparison. However, we think that this part could be improved and we will try to extend it in the new version.

**SC9 R#1:** The response of the model is estimated only from the differences between the "eclipse" and control simulations. However, the model performance can't be evaluated from real measurements, because the temporal resolution of the weather variables (air temperature and wind) in the BSRN stations is 3-hourly and thus they do not provide the required temporal resolution for an eclipse event. Given the density of climatic stations (especially in Europe), I was wondering on the availability of higher resolution meteorological data from neighbor (highly correlated) stations. A discussion about the daily (24-hour) variation of air temperature and wind speed/direction from SYNOP reports at BSRN stations and detection of possible differentiations between days before/during/after the eclipse should be of some value as well as regards the real response of surface layer at the sites of interest.

**A:** The set of ideas and proposals described in this comment are really valuable and interesting. A full response to this suggestion was provided in the Common comments and in **GC2 R#1**. From our understanding, this kind of study would be better for a future work. Nevertheless, we think that these ideas should appear in the manuscript because can be useful as a guideline for future analyses. Thus, we propose a new paragraph in the conclusions detailing the ideas for future works based on the method presented in this study and the code shared with NCAR.

Syntax and technical issues

**SC10 R#1:** Line 5 (page 1): avoid the phrase "..adding additional.."

**A:** This kind of sentences will be reworded. We have searched other parts of the text where it appears again.

**SC11 R#1:** Page 2, lines 2-4 , rephrase

[Figure]

**A:** We have reworded this sentence as

Original: For example, the region under the shadow of an eclipse experiences similar surface and planetary boundary layer (PBL) processes to those that occur at sunrise and sunset but they are more abruptly and on a shorter time-scale (Anderson, 1999). It provides a unique opportunity to analyze these processes.

New: Solar eclipse episodes are excellent experiments for analyzing the response of the atmosphere (e.g. surface and planetary boundary layer, PBL) and for testing the response of the physical schemes in NWP models. During a solar eclipse, the region under the shadow experiences a similar physical process that occurs at sunrise and sunset but abruptly and in a shorter time-scale (Anderson, 1999).

**SC12 R#1:** Page 3 , line 16. Leave space

**A:** : It is a typesetting error. We have solved it.

**SC13 R#1:** Page 5, line 12, , equation (3): t must be subscript, please correct

**A:** It is a typesetting error. It has been corrected, accordingly.

**SC14 R#1:** References of the same author should be put in chronological order (e.g Fernandez et al 1993a,b 1996).

**A:** You are right. We will check all references.

[Figure]

**SC15 R#1:** Some references are wrong or incomplete, e.g.: Founda and Melas (2007) should be cited as Founda et al. (2007) in the text and as following in the reference list : Founda, D., Melas, D., Lykoudis, S., Lisaridis, I., Gerasopoulos, E.,Kouvarakis, G., Petrakis, M., and Zerefos, C.: The effect of the total solar eclipse of 29 March 2006 on meteorological variables in Greece, Atmos. Chem. Phys., 7, 5543-5553, doi:10.5194/acp- 7-5543-2007, 2007.
Also, Montorne (2015) and Fernandez 1993b seem to be wrong or incomplete in the text and reference list.

**A:** It seems some problem or mistake that we did using Latex because we have checked our Bibtex file and all seems correct. We will fix this issue before sharing the files.

**SC16 R#1:** Figure 1, x-axis : labels appearance should be improved

**A:** We agree that the appearance of the labels in the x-axis of Fig. 1 is not appropriate. The idea was to include all the ticks for a better identification of the eclipses. We will reduce the number of labels for a better appearance. You can see the new version in Fig. 1.

**SC17 R#1:** Page 9 lines..1-10. The presentation of these results are somewhat confusing. Phrases like..'the worst europena site..' must be avoided, please rephrase.

**A:** Certainly the presentation of results in this paragraph is a little confusing. In the new version of the manuscript we have rewritten it. Moreover, we agree that this kind of sentences are awkward and they must be avoided. We will rephrase the paragraph as

"Sites under cloudless conditions show the highest improvements in MAE (Table 2). In

FLO, the use of WRF-eclipse represents an improvement of 90% in the MAE regarding WRF3.6.1 simulations. TAM shows similar results with a decrease of 77% in the MAE. Both sites show a high reduction of the bias. In FLO, from 438 to -34 $Wm^{-2}$ while in TAM 348 to -82 $Wm^{-2}$. This high underestimation is associated to the near grid-point issue mentioned before.

In contrast, lower improvement is observed in cloudy conditions. The best results are detected in CAR with an enhancement of 86%, drifting from a high positive bias of 364 $Wm^{-2}$ to a slightly negative one of -42 $Wm^{-2}$. LIN and PAY show similar improvement of 73% and 71% in the MAE, respectively. Finally, the minor improvement is observed in the Asian stations with variations in the MAE of +50% in TAT and +64% in XIA. The bias drifts from 493 to 176 $Wm^{-2}$ in XIA and from 798 to 395 $Wm^{-2}$ in TAT."

**SC18 R#1:** Fig 4: GH is not included in station PAY?

**A:** PAY is a site located near a lake named Lake Neuchâtel. Due to the coarse resolution used for these experiments, the nearest grid-point for this site is located over the lake (i.e. water body). In water bodies, the GH is 0 in the control simulation and in the new implementation. Therefore, as the differences are zero in both cases, the line was not included for this site. This issue is explained in the text (page 10) but probably it should be included in the caption of Fig. 4 in order to avoid any confusion. The answer to this question is related with **SC18 R#2**.

[Figure]

**Fig. 1.** New version of Fig. 1 in the manuscript

---

## Author Comment (AC2) · 10 Mar 2016

**Response to "Reviewer Comments", Anonymous Referee #2**

We are thankful to Anonymous Referee #2 for his, comments, suggestions and ideas. Without any doubt whatsoever, all of them are valuable to improve the quality of our study. In the current document, we will answer each one with more emphasis to those considerations that deal with technical or scientific considerations than those related to the language aspects.

We have decided to give a personalized reply to each referee. However, some points are common for both or they have a full impact on the entire paper. For this reason, we will present firstly a common block and then we will discuss each review point by point.

Hereinafter, we will use R#1 and R#2 as Referee #1 and #2, respectively.

In order to contextualize the response, the referee's commentary appears before our answer. Each review is quoted in gray. Our response appears with A: (from Authors) at the beginning and in black color. Each one is identified with a label composed by a number and a tag: GC (General Comments) and SC (Specific Comments). For example, SC4 refers to the 4th specific comment. During the discussion, the reader can find some cross-references between responses for R#1 and R#2. For example, **SC7 R#1** means the 7th specific comment of R#1.

Some answers that are also addressed to the Editor when we defend our position but we also think that the position of the reviewer is interesting. These responses are: **SC2 R#1**, **SC7 R#1**, **SC4 R#2**.

Regarding the submission of the revised manuscript, we wait until the final Editor's decision. At that moment, we will finish the last modifications and updates and we will submit the new version.

Common comments

**A:** The main idea behind the study presented in this manuscript was to discuss a new package for the WRF model (extensible to other General Circulation Models in the future) capable of representing any solar eclipse for any configuration in terms of date, domain size, grid resolution or projection, among others.

Under this framework, the manuscript was divided in three parts. The first part describes the implementation of the Bessel's method in the WRF model (Sect. 2) and it includes a validation of the algorithm by comparing our eclipse trajectories and the NASA's data-set (Sect. 3). The second part includes a validation of four real study cases for the GHI that is the most directly affected variable. The selection of these episodes was based on the availability of BSRN stations. This analysis has two major

goals: i) to provide a complementary validation of the algorithm for demonstrating that the degree of obscuration is well determined and it correlates in time with the real data and ii) to show the potentiality of this method for solar energy forecasting applications (Sec. 5.1). Finally, the last part of the manuscript introduces a brief description of the model response to show the applicability of this package for future academia research (Sec. 5.2).

The idea of including solar eclipses in the model born from the last partial and total solar eclipses that occurred in USA (October 23, 2014) and Europe (March 20, 2015), respectively, showing the necessity to incorporate these events into the solar parameterizations for solar renewable energy industry.

The main scope of this manuscript is the validation of the eclipse implementation by means of the trajectory and the GHI. From our understanding, some of the proposals of the referees for extending the study to the meteorological fields are very valuable as well as interesting but they are more appropriated as a future work. By including these extended analyses in the current version may blur the main flow of the study and may lead to a very large text.

The code used in this study has been shared with NCAR and we expect that it will be included in the next release (April, 2016). Consequently, the scientific community will be able to use this algorithm for a deep analysis of the model response and comparing with real measurements.

Further on the proposals for an extended study, both referees agree in the following aspects:

i) They suggested that it is necessary a different approach for the abstract (**SC1 R#1** and **SC1 R#2**) by including the main findings and conclusions as well as some qualitative results. We completely agree with this idea and we will rewrite the abstract. Following the idea provided by **SC1 R#2**, we have modified the last paragraph as:

"This contribution is divided in three parts. First, the implementation of the Bessel's method is validated for solar eclipses in the period 1950-2050, by comparing the shadow trajectory with values provided by NASA. Latitude and longitude are determined with a bias lower than 5 $10^{-3}$ degrees (i.e., ~550 m at Equator) being slightly overestimated and underestimated, respectively. The second part includes a validation of the simulated Global Horizontal Irradiance (GHI) for four total solar eclipses with measurements of the Baseline Surface Radiation Network (BSRN). The results show an improvement in MAE from 77% to 90% under cloudless skies. Lower agreement between modeled and measured GHI is observed under cloudy conditions since the effect of clouds is not considered in the radiative transfer schemes of the simulations. Finally, an introductory discussion of the response of meteorological variables (e.g. temperature, wind speed) to the reduction of GHI and shortwave heating rate is provided by comparing WRF-eclipse outcomes with control simulations."

ii) In **GC2 R#1** and **GC1 R#2** both referees indicated that they miss some comparison of the meteorological fields analyzed in Sect. 5.2 with real measurements. We agree with them that this kind of analysis would be interesting. Nevertheless, from our understanding, the best approach is to focus the current manuscript on the implementation of the method and prepare a future study for a better understanding of the atmosphere response with a large number of episodes and comparing with surface and vertical profile measurements.

iii) They indicated that Fig. 1 should be improved (**SC16 R#1** and **SC8 R#2**). We agree that this figure is awkward and useless with the current presentation. We have changed it as indicated in Fig. 1 of this document. We are opened to include new suggestions if necessary.

**Response to general comments**

**GC1 R#2:** Adding solar eclipse parameterizations to NWP models is not new, however, as pointed out by the authors, previous approaches were lacking generality and were usually designed for a particular case, only. The authors use the Bessel's method for the first time in a state-of-the-art NWP model and evaluate its performance. This work is interesting in the light of a growing demand for operational solar radiation variation forecasts (by the solar energy industry), which requires a general approach like the one described here. The paper is well structured and written. The model validation, however, should be improved. No comparison with real measurements regarding the surface layer response is included and only an idealized model setup was used (no cloud-radiation interaction and very coarse horizontal resolution). The paper can be considered publication after the following points have been addressed.

**A:** We appreciate the considerations of R#2 regarding our manuscript. As we indicated in Common comments, we agree that the comparison of the surface fields with real measurements is quite valuable but, in our opinion, this type of validation is more appropriate for a future manuscript with a different scope and based on the current one.

The reason for disabling the cloud-radiation interaction and the coarse grid resolution will be argued in **SC5 R#2**.

**Response to specific comments**

Abstract

**SC1 R#2:** Apart from providing a basic introduction into the topic, the abstract should focus on results/main findings. The latter is essentially lacking in the abstract. My

suggestion would be to replace or extend the last section of the abstract (line 16 - 21, page 1) by the main findings. Currently lines 16-21 are basically a repetition of the introductory lines 17-22 of page 4.

**A:** This suggestion and **SC1 R#1** indicate that it is necessary a new abstract improving the weak points that you have observed. You can see the new version in Common comments.

Implementation in the WRF-ARW model

**SC2 R#2:** A schematic representation of besselians elements showing the outline of umbra during eclipse on Earth's surface and its projection on the fundamental plane would be helpful.

**A:** Generally, the previous approaches used pre-computed solar eclipses or some kind of parameterization based on the eclipse track. The set of values used in the computations were provided by NASA catalogs and hence, they were indirectly based on the Bessel's method.

The advantage of our method with respect to the previous ones is that we incorporate one part of the Bessel's approach (Appendix A) inside the model in terms of the besselian elements. Consequently, the eclipse is evaluated during the simulation and thus it is independent of the grid size, resolution, projection and initialization.

A complementary response to this comment can be found in **GC1 R#1**.

**SC3 R#2:** Please elaborate a bit more on technical details of the implementation: what is the overhead/cost for this parameterization, including the setup phase (file read) in percentage of wall clock time? What is the size of the file containing the besselian elements which is read by the model? Is it read once, or opened and (partly) read at

**A:** In the following lines we will provide the technical details of the implementation. We have included a new module called module_ra_eclipse composed by three routines,

| | |
|---|---|
| solar_eclipse | main, |
| load_besselian_elements | search the besselian elements in run/eclipse_besselian_elements.dat, |
| compute_besselian_t | compute the besselian elements for a given time. |

Moreover, we include five new variables in the Registry:

| | |
|---|---|
| ra_sw_eclipse | namelist (physics) variable for enabling (1) and disabling (0, default) solar eclipses. No domain dependent |
| ECOBSC | history 2D variable representing the degree of obscuration at each grid-point |
| ECMASK | history 2D variable representing the status of the solar eclipse at each grid-point (0- No eclipse, 1- Partial/Penumbra region, 2-Total, 3- Annular) |
| elon_track, elat_track | coordinates of the path of the eclipse. |

At each call of the radiative transfer scheme, controlled by the radt variable, we check if the eclipse exists for that time step. We load the entire file called eclipse_besselian_elements.dat, every time and we check if the eclipse exists. If any episode exists then, we compute the eclipse conditions at each grid-point, if not, we come back to the main flow.

These processes do not require many computational time. In fact, when you compute the mean time for each time-step, the noise produced by the machine (i.e. other processes, programs, etc) has a higher effect than this new implementation.

The code is compatible with both cores: the ARW and the NMM. In the first case, with the solar schemes Dudhia, Goddard, New Goddard, CAM, RRTMG, RRTMG-fast and FLG and in the second one with the default scheme. This new implementation has been tested and shared with NCAR to be included in the next release.
Finally, the file eclipse_besselian_elements.dat is an ASCII file with a size of 4.5 kb. The processes of reading the file every time is not optimal from a programming point of view, but it is a recurrent approach in many parts of the model.

We are not really sure if this kind of information has to appear in the manuscript because it is not a technical report.

Algorithm validation

**SC4 R#2:** The authors mention, that there are some differences between the eclipse tracks computed within the WRF module and the NASA values. However, only very vague explanations are given like "associated with small differences on the code" and "truncation errors due to compiling options". Even though the differences are relatively small, they seem to be beyond simple truncation errors. To convince the reader of the correctness of the implementation, please point out the reasons for the observed differences more clearly. Do the differences decrease significantly when performing the computations in double precision as compared to single precision?

**A:** We agree that there are not many details regarding this point. The reason is because the errors are low for mesoscale applications and we considered that this kind of information was irrelevant for the reader.

We compared the computations by using double precision instead of single precision (Figs. 2 and 3). In this case, the bias in latitude becomes zero while in longitude is reduced by 5 (i.e. 110 m as a maximum).

The remaining differences in longitude evaluation are produced by two constants: i) the Earth's eccentricity and ii) the correcting factor for the true longitude considering the non-uniform rotation of the Earth (Eq. 40). Both constants are taken with single precision because we could not find data-sets with more precision. Moreover, by using double precision, the computational time increased and the improvement in accuracy is

not significant for most of the WRF applications in which solar eclipses can be enabled. Consequently, we decided to use single precision variables.

Finally, there are higher order effects as the compilation option or even the compiler that can lead to small differences in the results.

This information can be included in the manuscript, but from our perspective is not really relevant for most of the potential readers. We wait until the Editor's decision.

Results

**SC5 R#2:** Given the fact that even global models today use resolutions of O(10) km operationally and even higher resolutions in less time-critical scientific applications, the resolution chosen here seems rather coarse. One consequence of this coarse resolution is that the cloud interaction has been switched off, since reproducing the observed cloud structure is impossible anyway at this resolution. This approximation may be OK for qualitative comparisons against the measured GHI values and for evaluating the qualitative response of the WRF model as it is done here. The model setup chosen for validation is, however, very different from what would be used in real applications.

**A:** We completely agree with this statement. Having said this, from our perspective the used methodology is enough for the purposes of this study.

As we explained in the "Common Comments", the scope of this manuscript is to set the basis of a new method for studying solar eclipses with the WRF model. Therefore, we focus the paper on the method and on the validation of the algorithm instead of a deep analysis of the model response.

Consequently, as we are mainly interested in the representation of the eclipse but not in the accuracy of the meteorological fields, we created big domains with coarse resolution. The idea of creating domains covering a large area is to include the shadow during all the episode.We chose a coarse resolution for two reasons. Firstly, but ir-
relevant, because of the computational cost. Secondly, because we disable the cloud interaction in the radiative transfer and thus, the GHI varies slowly in the horizontal (i.e. homogeneous in the sub-grid).

Of course, this initial study can be improved with higher order modeling approaches such as enabling clouds, increasing the horizontal and vertical resolution or performing a sensitivity tests of the best options for each site. Nevertheless, we think that these ideas are more appropriate for future works because in this study may distort the main scope.

The response to this comment links with **SC6 R#2**.

**SC6 R#2:** The paper would benefit a lot from at least one fully fledged high-resolution run (i.e. with cloud-radiation interaction switched on) and comparison of the surface layer response against real measurements (surface temperature, wind, . . . ). One may e.g. choose the europe episode, run the model for a shrinked model domain over central europe and do a validation for Lindenberg for which high resolution data should be available.

**A:** This comment is directly linked with **SC5 R#2** and **SC9 R#1**. Certainly, we agree that this kind of analysis can be very valuable but, in our understanding, they are not appropriated for this study.

The main scope of the manuscript is the implementation of the Bessel's method within the WRF-ARW model and a validation of the algorithm but not a full study of the impact of solar eclipses in the atmosphere and the reliability of the models for modeling this response.

Therefore, these ideas can be developed in future studies more focused on the response of the atmosphere that can be based in the work proposed in this manuscript.

We propose to include a new paragraph incorporating all these ideas in the Conclusions.

Technical issues

**SC7 R#2:** Typo in Reference to Chauvenet et al. : Chavuenet -> Chauvenet.

**A:** It was a typesetting error. We have corrected the reference accordingly.

**SC8 R#2:** Fig. 1: looks a bit crowded due to the use of 6 different colors. It may improve when using only 3 different colors for A/H/T and 2 different line styles for lat/lon.

**A:** The answer to this comment links with **SC14 R#1**. We agree that this figure must be re-plotted. R#1 suggested a reduction of ticks in the x-axis. You can see the new Figure in Fig. 1.

**SC9 R#2:** Fig. 2: Please add some additional shading, indicating e.g. the totality zone or 90% obscuration area. This may help the unexperienced reader to assess more easily to what degree the various stations are affected by the eclipse.

**A:** This is an excellent idea. We will include a new version of Fig. 2 adding this information.

**SC10 R#2:** Table 1: please add the maximum degree of obscuration for each site.

**A:** This information appears in Fig. 3, but we will include this information in Table 1 to make the manuscript easier to read.
**SC11 R#2:** Page 9, line 29: superfluous "of"

**A:** Of course. We have removed the superfluous "of".

**SC12 R#2:** Page 11, line 27: superfluous "a" at "a near-zero . . .

**A:** Sure. The superfluous "a" has been removed.

**SC13 R#2:** Page 12, line 5 "the observer (i.e. position within the model domain)". Please add the description given brackets already at some previous occurrence of "observer" in order to better clarify what is meant by "observer".

**A:** We will provide this information earlier to make the text easier to read.

**SC14 R#2:** Page 12, line 9: "This validation show . . . " missing "s".

**A:** We agree. This part has been reworded accordingly.

**SC15 R#2:** Page 12, line 32: typo "shadowm"

**A:** It is typesetting issue. This word has been reworded.

Appendix B: Model description

Please add some information about

**SC16 R#2:** The source of the applied boundary data and its update frequency (so far

only the initialization is described).

**A:** We use the ERA Interim Reanalysis with an update frequency of 6 h (i.e. the available output of this model) as initial and boundary conditions. This information will be included in the Appendix B.

**SC17 R#2:** The time step.

**A:** We use an adaptive time-step, for this reason it was not included in the model configuration. The first guest is set to 30 s and we set a target CFL condition of 1.2. The time step can not increase more than 60 s because this is the output frequency of the history file. This kind of information will be included in Appendix B.

**SC18 R#2:** Since a lake-point was selected for comparison with the station PAY, please document whether a lake model/lake parameterization was used by WRF (i.e. is the lake (surface) temperature prognostic, or is it constant (like SST).

**A:** We use the default option, i.e. without lake model. As it is discussed in **SC5 R#2**, the initial idea of this manuscript was not to perform an accurate description of the variables at each site. The description of the treatment of the lake surface will be included in Appendix B as you requested.
* * *
[Figure]

**Fig. 1.** New version of Fig. 1 in the manuscript.

[Figure]

**Fig. 2.** Scatterplot comparing the errors in latitude and longitude considering single and double precision.

[Figure]

**Fig. 3.** Equivalent to Fig. 1 in the manuscript using double precision.